# Intrinsic timescales in the visual cortex change with selective attention and reflect spatial connectivity

Roxana Zeraati [1,2], Yan-Liang Shi [3,4], Nicholas A. Steinmetz [5], Marc A. Gieselmann[6], Alexander Thiele[6], Tirin Moore [7], Anna Levina [2,8,9,10] ✉ & Tatiana A. Engel [3,4,10] ✉

Intrinsic timescales characterize dynamics of endogenous fluctuations in neural activity. Variation of intrinsic timescales across the neocortex reflects functional specialization of cortical areas, but less is known about how intrinsic timescales change during cognitive tasks. We measured intrinsic timescales of local spiking activity within columns of area V4 in male monkeys performing spatial attention tasks. The ongoing spiking activity unfolded across at least two distinct timescales, fast and slow. The slow timescale increased when monkeys attended to the receptive fields location and correlated with reaction times. By evaluating predictions of several network models, we found that spatiotemporal correlations in V4 activity were best explained by the model in which multiple timescales arise from recurrent interactions shaped by spatially arranged connectivity, and attentional modulation of timescales results from an increase in the efficacy of recurrent interactions. Our results suggest that multiple timescales may arise from the spatial connectivity in the visual cortex and flexibly change with the cognitive state due to dynamic effective interactions between neurons.

The brain processes information and coordinates behavioral sequences over a wide range of timescales[1–3]. While sensory inputs can be processed as fast as tens of milliseconds[4–7], cognitive processes such as decision-making or working memory require integrating information over slower timescales from hundreds of milliseconds to minutes[8–10]. These differences are paralleled by the timescales of intrinsic fluctuations in neural activity across the hierarchy of cortical areas. The intrinsic timescales are defined by the exponential decay rate of the autocorrelation of activity fluctuations. The intrinsic timescales are faster in sensory areas, intermediate in association cortex, and slower in prefrontal cortical areas[11]. The hierarchy of intrinsic timescales is observed across different recording modalities including spiking activity[11,12], intracranial electrocorticography (ECoG)[13,14], and functional magnetic resonance imaging (fMRI)[15,16]. The hierarchy of intrinsic timescales reflects the specialization of cortical areas for behaviorally relevant computations, such as the processing of rapidly changing sensory inputs in lower cortical areas and long-term integration of information (e.g., for evidence accumulation, planning, etc.) in higher cortical areas[17].

[1]International Max Planck Research School for the Mechanisms of Mental Function and Dysfunction, University of Tübingen, Tübingen, Germany. [2]Max Planck Institute for Biological Cybernetics, Tübingen, Germany. [3]Cold Spring Harbor Laboratory, Cold Spring Harbor, NY, USA. [4]Princeton Neuroscience Institute, Princeton University, Princeton, NJ, USA. [5]Department of Biological Structure, University of Washington, Seattle, WA, USA. [6]Biosciences Institute, Newcastle University, Newcastle upon Tyne, UK. [7]Department of Neurobiology and Howard Hughes Medical Institute, Stanford University, Stanford, CA, USA. [8]Department of Computer Science, University of Tübingen, Tübingen, Germany. [9]Bernstein Center for Computational Neuroscience Tübingen, Tübingen, Germany. [10]These authors contributed equally: Anna Levina, Tatiana A. Engel. ✉e-mail: anna.levina@uni-tuebingen.de; tatiana.engel@princeton.edu

In addition to ongoing fluctuations characterized by intrinsic timescales, neural firing rates also change in response to sensory stimuli or behavioral task events. These stimulus or task-induced dynamics are characterized by the timescales of trial-average neural response[18,19] or encoding various task events over multiple trials[12,20]. The task-induced timescales also increase along the cortical hierarchy[12,14,20–22]. However, task-induced and intrinsic timescales are not correlated across individual neurons in any cortical area[12], suggesting they may arise from different mechanisms. Indeed, the timescales of trial-average response increase through the mouse visual cortical hierarchy, whereas the intrinsic timescales do not change[22]. Moreover, the task-induced and intrinsic timescales can depend differently on task conditions. For example, for a fixed trial-average response in a specific task condition, the intrinsic timescale of neural dynamics varies substantially across trials and these changes are predictive of the reaction time in a decision-making task[23]. While task-induced timescales relate directly to task execution, less is known about how intrinsic timescales change during cognitive tasks. Intrinsic timescales measured with ECoG exhibit a widespread increase across multiple cortical association areas during working memory maintenance, consistent with the emergence of persistent activity in this period[13]. However, whether intrinsic timescales can change with temporal and spatial specificity in local neural populations processing specific information during a task has not been tested. It is also unclear whether intrinsic timescales can flexibly change in sensory cortical areas and in cognitive processes other than memory maintenance.

The mechanism underlying the diversity of intrinsic timescales across cortical areas can be related to differences in the connectivity. The hierarchical organization of timescales correlates with the gradients in the strength of neural connections in different cortical areas[24,25]. These gradients exhibit an increase through the cortical hierarchy in the spine density on dendritic trees of pyramidal neurons[26,27], gray matter myelination[13,28], expression of N-methyl-D-aspartate (NMDA) and gamma-aminobutyric acid (GABA) receptor genes[13,29], strength of structural connectivity measured using diffusion MRI[16], or strength of functional connectivity[15,16,30–32].

The relation between the connectivity and timescales is further supported by computational models. Differences in timescales across cortical areas can arise in network models from differences in the strength of recurrent excitatory connections[27,33]. These models matched the strength of excitatory connections to the spine density of pyramidal neurons[27] or to the strength of structural connectivity[33] in different cortical areas. Moreover, models demonstrate that the topology of connections in addition to the connection strength can affect the timescales of network dynamics. For example, slower timescales emerge in networks with clustered connections compared to random networks[34], or heterogeneity in the strength of inter-node connections gives rise to diverse localized timescales in a one dimensional network[35]. Thus, network models can relate dynamics to connectivity and generate testable predictions to identify mechanisms underlying the generation of intrinsic timescales in the brain.

We examined how the intrinsic timescales of spiking activity in visual cortex were affected by the trial-to-trial alterations in the cognitive state due to visual spatial attention. We analyzed spiking activity recorded from local neural populations within cortical columns in primate area V4 during two different spatial attention tasks and a fixation task. In all tasks, the autocorrelation of intrinsic activity fluctuations showed at least two distinct timescales, one fast and one slow. The slow timescale was longer on trials when monkeys attended to the receptive fields of the recorded neurons and correlated with the monkeys' reaction times. We used recurrent network models to test several alternative mechanisms for generating the multiplicity of timescales and their flexible modulation. We established analytically that spatially arranged connectivity generates multiple timescales in local population activity and found support for this theoretical

prediction in our V4 recordings. In contrast, heterogeneous biophysical properties of individual neurons alone cannot account for both temporal and spatial structure of V4 correlations. Thus, the V4 timescales arise from spatiotemporal population dynamics shaped by the local spatial connectivity structure. The model indicates that modulation of timescales during attention can be explained by a slight increase in the efficacy of recurrent interactions. Our results suggest that multiple intrinsic timescales in local population activity arise from the spatial network structure of the neocortex and the slow timescales can flexibly adapt to trial-to-trial changes in the cognitive state due to dynamic effective interactions between neurons.

## Results
### Multiple timescales in fluctuations of local neural population activity

We analyzed spiking activity of local neural populations within cortical columns of visual area V4 from monkeys performing a fixation task (FT) and two different spatial attention tasks (AT1, AT2)[36,37] (Fig. 1a–c, Supplementary Fig. 1). The activity was recorded with 16-channel linear array microelectrodes from vertically aligned neurons across all cortical layers such that the receptive fields (RFs) of neurons on all channels largely overlapped. In FT, the monkey was rewarded for fixating on a blank screen for 3 s on each trial (Fig. 1a). During AT1, the monkeys were trained to detect changes in the orientation of a grating stimulus in the presence of three distractor stimuli and to report the change with a saccade to the opposite location (antisaccade, Fig. 1b). On each trial, a cue indicated the stimulus that was most likely to change, which was the target of covert attention, and the stimulus opposite to the cue was the target of overt attention due to the antisaccade preparation. During AT2, the monkey was rewarded for detecting a small luminance change in a grating stimulus in the presence of a distractor stimulus placed in the opposite hemifield. The monkey reported the change by releasing a bar. An attentional cue on each trial indicated the stimulus where the change should be detected, which was the target of covert attention (Fig. 1c).

We analyzed the timescales of fluctuations in local spiking activity by computing the autocorrelations (ACs) of spike counts in 2 ms bins. Previous laminar recordings showed that the neural activity is synchronized across cortical layers alternating spontaneously between synchronous phases of high and low firing rates[36,38]. Therefore, we pooled the spiking activity across all layers (Fig. 1d) to obtain more accurate estimates of the spike-count autocorrelations. The shape of spike-count autocorrelations in our data deviated from a single exponential decay. In logarithmic-linear coordinates, the exponential decay corresponds to a straight line with a constant slope. The spike-count autocorrelations exhibited more than one linear slope, with a steep initial slope followed by shallower slopes at longer lags (Fig. 1e). The multiple decay rates in the autocorrelations indicate the presence of multiple timescales in the fluctuations of local population spiking activity.

To verify the presence of multiple timescales and to accurately estimate their values from autocorrelations, we used a method based on adaptive Approximate Bayesian Computations (aABC, Methods)[39]. This method overcomes the statistical bias in autocorrelations of finite data samples, which undermines the accuracy of conventional methods based on direct fitting of the autocorrelation with exponential decay functions. The aABC method estimates the timescales by fitting the spike-count autocorrelation with a generative model that can have single or multiple timescales and incorporates spiking noise. The method accounts for the finite data amount, non-Poisson statistics of the spiking noise, and differences in the mean and variance of firing rates across experimental conditions. The aABC method returns a posterior distribution of timescales that quantifies the estimation uncertainty and allows us to compare alternative hypotheses about the number of timescales in the data.

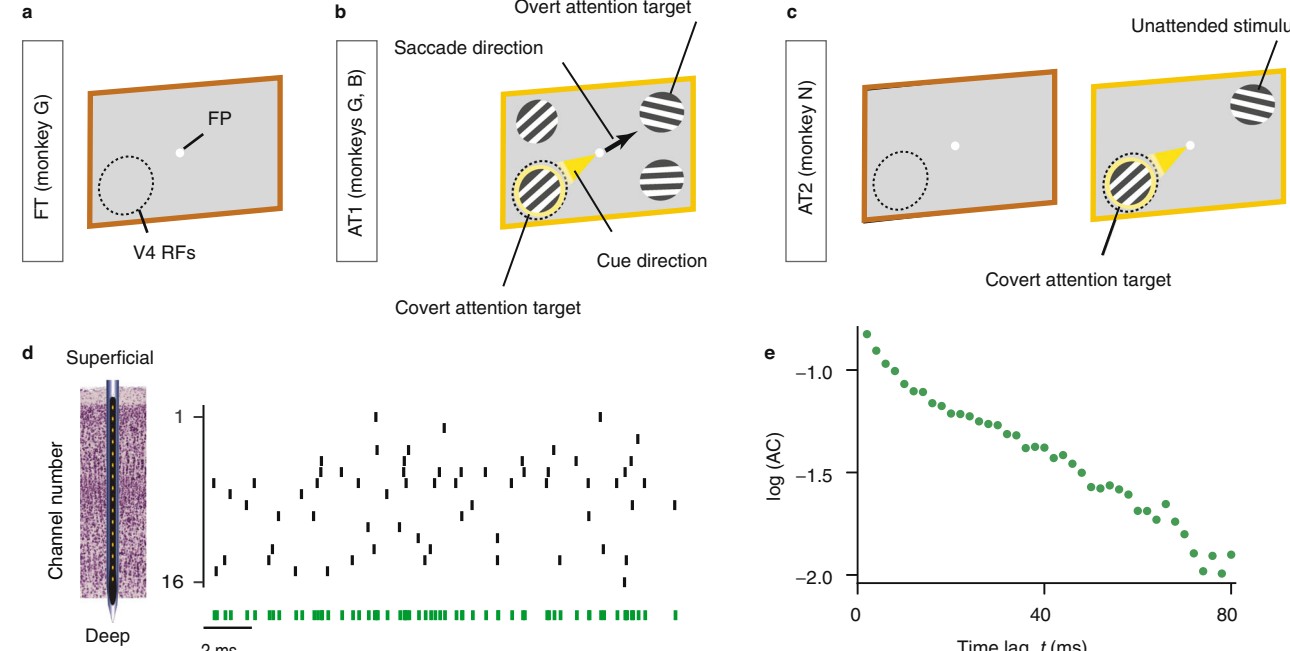

**Fig. 1 | Computing autocorrelations of spiking activity in V4 columns during fixation and attention tasks. a** In the fixation task (FT), the monkey was rewarded for fixating a central fixation point (FP) on a blank screen for 3 s on each trial. **b** In the attention task 1 (AT1), monkeys were trained to detect an orientation change in one of four peripheral grating stimuli, while an attention cue indicated which stimulus was likely to change. Monkeys reported the change with a saccade to the stimulus opposite to the change (black arrow). The cued stimulus was the target of covert attention (yellow spotlight), while the stimulus opposite to the cue was the target of overt attention. **c** In the attention task 2 (AT2), the monkey was rewarded for detecting a small luminance change in one of two grating stimuli, directed by an attention cue. The monkey responded by releasing a bar. The brown frame shows the blank screen in the pre-stimulus period. In all tasks, epochs marked with brown frames were used for analyses of spontaneous activity and epochs marked with orange frames were used for the analyses of stimulus-driven activity. The cue was either a short line (AT1) or two small dots (AT2) indicating the covert attention target. The dashed circle denotes the receptive field locations of recorded neurons (V4 RFs) and was not visible to the monkeys (see Supplementary Fig. 1 for details). **d** Multi-unit spiking activity (black vertical ticks) was simultaneously recorded across all cortical layers with a 16-channel linear array microelectrode. The autocorrelation of spike counts in 2 ms bins was computed from the spikes pooled across all channels (green ticks). **e** The autocorrelation (AC) computed from the pooled spikes on an example recording session. Multiple slopes visible in the autocorrelation in the logarithmic-linear coordinates indicate multiple timescales in neural dynamics. Source data for panels **d** and **e** are provided as a Source Data file.

We fitted each autocorrelation with a one-timescale ($M_1$) and a two-timescale ($M_2$) generative model and selected the optimal number of timescales by approximating the Bayes factor obtained from the posterior distributions of the fitted models (Fig. 2a, Supplementary Fig. 2, Methods). The majority of autocorrelations were better described by the model with two distinct timescales ($M_2$) than with the one-timescale model (Fig. 2a, b). The presence of two distinct timescales (fast $\tau_1$ and slow $\tau_2$) was consistent across both spontaneous (i.e. in the absence of visual stimuli, $\tau_{1,MAP} = 8.87 \pm 0.78$ ms, $\tau_{2,MAP} = 85.82 \pm 15.9$ ms, mean ± s.e.m. across sessions, MAP: Maximum *a posteriori* estimate from the multivariate posterior distribution) and stimulus-driven activity ($\tau_{1,MAP} = 5.05 \pm 0.51$ ms, $\tau_{2,MAP} = 135.87 \pm 9.35$ ms, mean ± s.e.m.), and across all monkeys, while the precise values of timescales were heterogeneous reflecting subject- or session-specific characteristics (Fig. 2c). Although it is possible that autocorrelations contained more than two timescales, with our data amount, the three-timescale model did not provide a better fit than the two-timescale model (Supplementary Fig. 3). Thus, the two-timescale model provided a parsimonious description of neural dynamics in our data.

**Slow timescales are modulated during spatial attention**

Next, we examined whether the intrinsic timescales of spiking activity were modulated during spatial attention. We compared the timescales estimated from the stimulus-driven activity on trials when the monkeys attended toward the RFs location of the recorded neurons (attend-in condition, covert or overt) versus the trials when they attended outside the RFs location (attend-away condition). In this analysis, we included recording sessions in which the autocorrelations were better fitted with two timescales in both attend-away and attend-in (covert or overt) conditions. We compared the MAP estimates of the fast $\tau_1$ and slow $\tau_2$ timescales between attend-in and attend-away conditions across recording sessions.

We found that the slow timescale was significantly longer during both covert and overt attention relative to the attend-away condition (covert: mean $\tau_{2,att-in} = 140.69$ ms, mean $\tau_{2,att-away} = 115.07$ ms, $p = 3 \times 10^{-4}$, $N = 32$; overt: mean $\tau_{2,att-in} = 141.31$ ms, mean $\tau_{2,att-away} = 119.58$ ms, $p = 7 \times 10^{-4}$, $N = 26$; two-sided Wilcoxon signed-rank test) (Fig. 3), while there was no significant change in the fast timescale during attention (covert: mean $\tau_{1,att-in} = 5.53$ ms, mean $\tau_{1,att-away} = 5.54$ ms, $p = 0.75$, $N = 32$; overt: mean $\tau_{1,att-in} = 3.42$ ms, mean $\tau_{1,att-away} = 4.12$ ms, $p = 0.39$, $N = 26$; two-sided Wilcoxon signed-rank test). The increase in the slow timescale with attention was evident on individual recording sessions when comparing the marginal posterior distributions of $\tau_2$ for attend-in versus attend-away conditions (Fig. 3a, d). The significant increase of $\tau_2$ was observed in 24 out of 32 individual sessions during covert attention, and 22 out of 26 individual sessions during overt attention. Both fast and slow timescales varied across sessions, but were not significantly different between covert and overt attention ($p > 0.05$ for both $\tau_1$ and $\tau_2$, two-sided Wilcoxon signed-rank test, Supplementary Fig. 4). The increase in $\tau_2$ was not due to increase in the firing rate with attention, since the aABC method accounts for the differences in the firing rate across behavioral conditions (Methods), and $\tau_2$ was not correlated with the mean firing rate of population activity (Supplementary Fig. 5). The increase of slow timescales during attention is consistent with the reduction in the power of low-frequency fluctuations in local field potentials[37,40–42] and spiking activity[43] (Supplementary

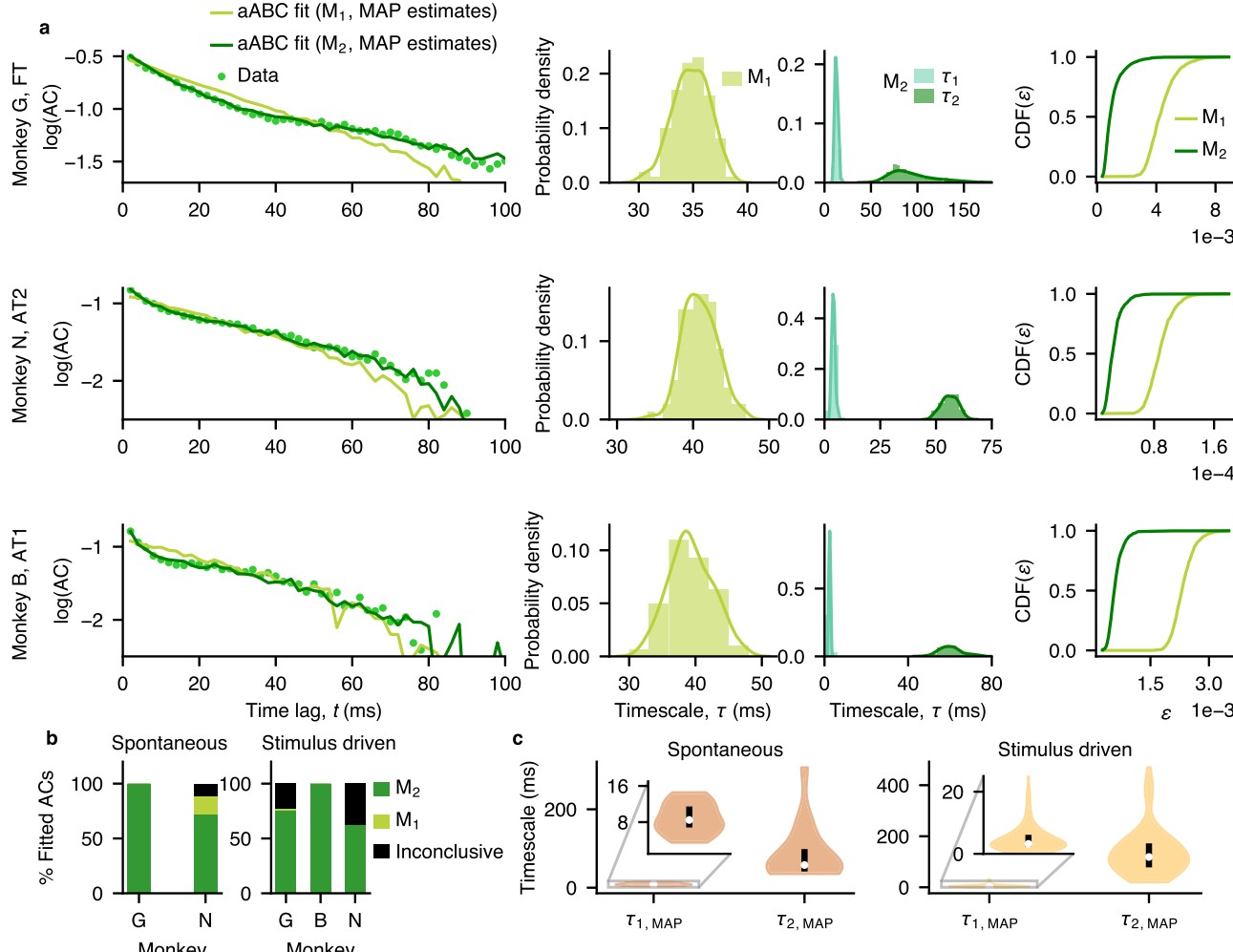

**Fig. 2 | Two timescales in ongoing spiking activity within V4 columns.**
**a** Comparison between the two-timescale ($M_2$) and one-timescale ($M_1$) generative models for three example recording sessions (rows). The models were fitted to autocorrelations of V4 spiking activity using the adaptive Approximate Bayesian Computations (aABC). The shape of the neural autocorrelation (AC) is reproduced by the autocorrelation of synthetic data from the two-timescale model with the maximum *a posteriori* (MAP) parameters, but not by the one-timescale model (left panels). Autocorrelations are plotted from the first time-lag ($t = 2$ ms). Marginal posterior distribution of the timescale ($\tau$) estimated by fitting $M_1$ is in between the posterior distributions of timescales ($\tau_1$, $\tau_2$) estimated by fitting $M_2$ (middle panels). Cumulative distribution of errors $CDF_{M_i}(\varepsilon)$ between the autocorrelations of V4 data and synthetic data generated with parameters sampled from the $M_1$ or $M_2$ posteriors (right panels). $M_2$ is a better fit since it produces smaller errors (i.e. Bayes factor = $CDF_{M_2}(\varepsilon)/CDF_{M_1}(\varepsilon) > 1$, Methods). FT - fixation task, AT - attention task. **b** In most recording sessions, the autocorrelations during spontaneous and stimulus-

driven activity were better described with two distinct timescales ($M_2$) than a single timescale ($M_1$). For a few fits the model comparison was inconclusive as the observed statistics were insufficient to distinguish between the models. The total number of fitted autocorrelations for each monkey (G, N, B) was $N_G = 5$, $N_N = 18$ for spontaneous, and $N_G = 57$, $N_N = 24$, $N_B = 39$ for stimulus-driven activity. **c** MAP estimates for the fast and slow timescales were heterogeneous across recording sessions during spontaneous and stimulus-driven activity. Violin plots show the distributions of timescales for the autocorrelations that were better fitted with two timescales. The distributions were smoothed with Gaussian kernel densities. The white dot indicates the median, the black box is the first to third quartiles. Inset shows a zoomed range for the fast timescale. Number of autocorrelations better fitted with $M_2$ out of the total fitted autocorrelations: $N = 18/23$ during spontaneous activity, $N = 92/120$ during stimulus-driven activity. Source data are provided as a Source Data file.

Note 1, Supplementary Fig. 6, 7). The modulation of the slow timescale was consistent across both attention tasks (AT1 and AT2) and each monkey, and appeared in response to trial-to-trial changes in the cognitive state of the animal directed by the attention cue. These results suggest that different mechanisms control the fast and slow timescales of ongoing spiking activity, and the mechanisms underlying the slow timescale can flexibly adapt according to the animal's behavioral state.

To test whether attentional modulation of timescales was relevant for behavior, we analyzed the relationship between timescales and monkeys' reaction times in the attention tasks. We quantified the relationship between the average reaction times of monkeys' responses in each session (see Supplementary Fig. 1 for details of experiment) and the MAP estimated timescales of spiking activity

using linear mixed-effects models fitted separately in attend-in and attend-away conditions (Fig. 4, Methods, Supplementary Table 1, 2). The linear mixed-effects models had a separate intercept for each monkey to account for individual differences between the monkeys and attention tasks (AT1 and AT2). The reaction times were negatively correlated with the slow timescales in attend-in condition (combined covert and overt) (slope = $-0.16 \pm 0.066$, mean ± 95% confidence intervals (CIs); $p = 9 \times 10^{-6}$, F-test; $N = 58$, $R^2 = 0.62$), but not in attend-away condition (slope = $0.015 \pm 0.12$, $p = 0.79$, $N = 32$, $R^2 = 0.69$). Fast timescales were not correlated with the reaction times (attend-in: slope = $0.0016 \pm 0.86$, $p = 0.997$, $N = 58$, $R^2 = 0.46$; attend-away: slope = $0.53 \pm 0.94$, $p = 0.26$, $N = 32$, $R^2 = 0.70$). Thus, on average monkeys responded to a stimulus change faster in sessions

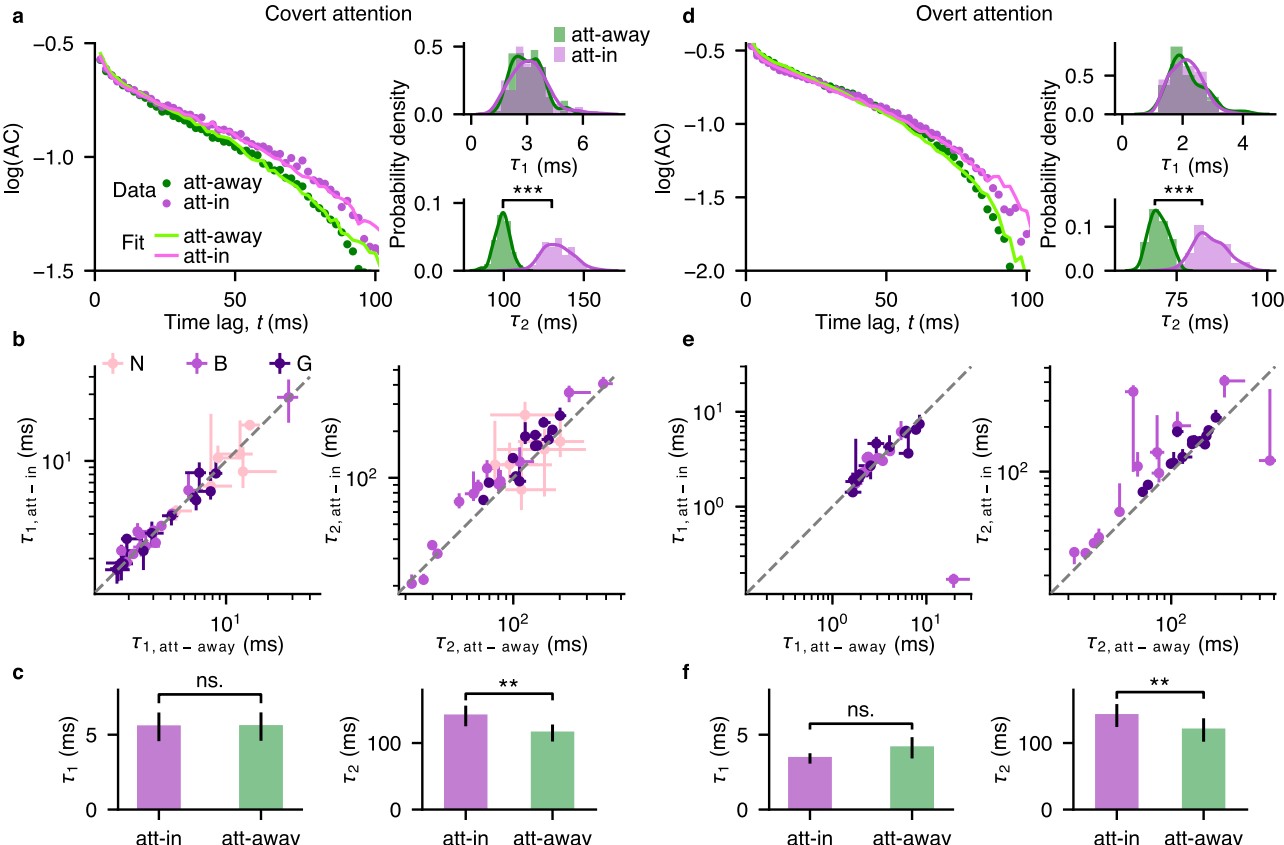

**Fig. 3 | Slow timescales increase during spatial attention. a** Autocorrelations (ACs) of neural data with two-timescale fits (left) and the corresponding posterior distributions (right) during covert attention and attend-away condition for an example recording session. The fitted lines are autocorrelations of synthetic data from the two-timescale model with MAP parameters. The posterior distribution of the slow timescale ($\tau_2$) has significantly larger values in attend-in (att-in) than in attend-away (att-away) condition. Statistics: two-sided Wilcoxon rank-sum test, $p_{\tau_1} = 0.43$, $p_{\tau_2} < 10^{-10}$. Number of samples in each posterior $N = 100$. **b** The increase of the slow timescale ($\tau_2$, right) during attention was visible on most sessions (points - MAP estimates for individual sessions, error bars - the first and third quartiles of the marginal posterior distribution, dashed line - the unity line). If the MAP estimate was smaller than the first or larger than the third quartile, the error bar was discarded. Larger error bars indicate wider posteriors, i.e. larger estimation

uncertainty. Number of included sessions (in which autocorrelations were better fitted with M₂) from the total fitted sessions for each monkey: $N_G = 13/19$, $N_B = 13/13$, $N_N = 6/12$. Color of the dots indicates different monkeys. **c** Across sessions, the fast timescale ($\tau_1$, left) did not change, while the slow timescale ($\tau_2$, right) significantly increased during covert attention (magenta) relative to the attend-away condition (green). Bar plots show the mean ± s.e.m of MAP estimates across sessions. Statistics: two-sided Wilcoxon signed-rank test, $p_{\tau_1} = 0.75$, $p_{\tau_2} = 3 \times 10^{-4}$. ns., **, *** indicate $p > 0.05/4$, $p < 10^{-2}$, $p < 10^{-3}$, respectively (Bonferroni corrected for 4 comparisons). Individual data points are shown in **b. d–f** Same as **a–c** but during the overt attention for a different example session. Number of included sessions from the total fitted sessions for each monkey: $N_G = 14/19$, $N_B = 12/12$. Statistics: **d** $p_{\tau_1} = 0.07$, $p_{\tau_2} < 10^{-10}$, (**f**) $p_{\tau_1} = 0.39$, $p_{\tau_2} = 7 \times 10^{-4}$. Source data are provided as a Source Data file.

with longer slow timescales of neurons with the receptive fields in the attended location. The spatial selectivity of this effect suggests that the increase in the slow timescale may be related to behavioral benefits of selective spatial attention.

## Mechanisms for generating multiple timescales in local population dynamics

What mechanisms can generate multiple timescales in the local population activity? One possibility is that multiple timescales reflect biophysical properties of individual neurons within a local population. For example, two timescales can arise from mixing heterogeneous timescales of different neurons[44,45] or combining different biophysical processes, such as a fast membrane time constant and a slow synaptic time constant[46]. Alternatively, multiple timescales in local population activity can arise from spatiotemporal population dynamics in networks with spatially arranged connectivity[47].

Analyses of well-isolated single-unit activity (SUA) would be ideal for testing whether multiple timescales in local V4 population activity reflect the mixing of heterogeneous timescales of individual neurons or dynamics shared by the population. However, due to low

firing rates, SUA did not yield sufficient data for conclusive model comparison. We fitted autocorrelations of SUA during the fixation task (which had the longest trial duration of 3 s and thus the largest data amount per trial) and performed the model comparison to determine the number of timescales. While some single units clearly showed two distinct timescales, the model comparison was inconclusive for most units because autocorrelations were dominated by noise due to low data amount (Supplementary Note 2, Supplementary Fig. 8). We, therefore, turned to computational modeling for testing possible alternative mechanisms for generating multiple timescales.

To determine which mechanism, local biophysical properties or spatial network interactions, is consistent with neural dynamics in V4, we developed three recurrent network models each with a different mechanism for timescale generation (Fig. 5). We implemented all mechanisms within the same modeling framework. The models consist of binary units arranged on a two-dimensional lattice corresponding to lateral dimensions in the cortex (Fig. 5a–c). Each unit represents a small population of neurons, such as a cortical minicolumn[48,49], and is connected to 8 other units in the network. The activity of unit $i$ at time-step

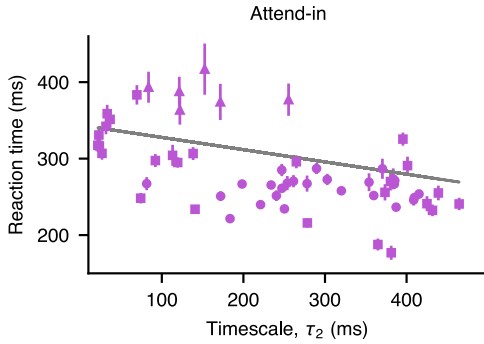
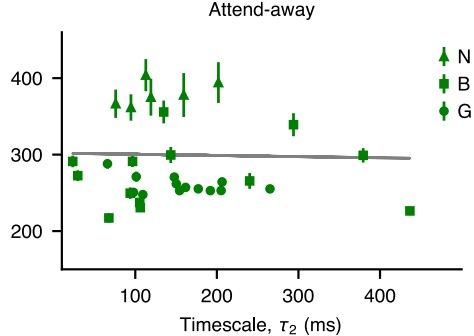

**Fig. 4 | Slow timescales are correlated with monkeys' reaction times.** Average reaction times of monkeys for each session were negatively correlated with the MAP estimates of slow timescales ($\tau_2$) in attend-in condition (left, slope = $-0.16 \pm 0.066$, mean $\pm$ 95% CIs, $p = 9 \times 10^{-6}$, one-sided F-test, $N = 58$, $R^2 = 0.62$) but not attend-away condition (right, slope = $0.015 \pm 0.12$, $p = 0.79$, one-sided $F$-test, $N = 32$, $R^2 = 0.69$). Each point represents one recording session; symbols indicate different monkeys. Error bars denote $\pm$ s.e.m. Gray lines show the estimated fixed-effect parameters (slope and intercept) of the fitted mixed-effects model (Methods, Supplementary Table 1). Because reaction time was tested against 4 hypothesized predictors (two timescales in two attention conditions), a Bonferroni-adjusted significance level was calculated as $p < 0.05/4$. Source data are provided as a Source Data file.

$t'$ is described by a binary variable $S_i(t') \in \{0,1\}$ representing high (1) and low (0) firing-rate states of a local population[36]. The activity $S_i(t')$ stochastically transitions between states driven by the self-excitation (probability $p_s$), excitation from the connected units (probability $p_r$), and the stochastic external excitation (probability $p_{ext} \ll 1$) delivered to each unit (Methods). The self-excitation probability describes intrinsic dynamics of a unit in the absence of network interactions, arising from biophysical properties of neurons or reverberation within a local population (via the vertical connectivity within a minicolumn). The self-excitation generates a timescale $\tau_{self}$, which is the autocorrelation timescale of a two-state Markov process: $\tau_{self} = (-\ln(p_s))^{-1}$ (Methods, Supplementary Note 3). The recurrent excitation $p_r$ accounts for horizontal interactions between units. The sum of all interaction probabilities is the local branching parameter: $BP = p_s + 8p_r$, describing the expected number of units activated by a single active unit $i$.

The models differ in the mechanism generating multiple timescales in the local population activity. In two models, connectivity is random and multiple timescales arise locally from biophysical properties of individual units. In the third model, connectivity is spatially organized and multiple timescales arise from recurrent interactions between units[47].

The first model assumes that two timescales in local population activity reflect aggregated activity of different neuron types with distinct (fast and slow) biophysical timescales (e.g., membrane time constants), which we modeled as two types of units (A and B) each with a different self-excitation probability ($p_{s,A}$, $p_{s,B}$, Fig. 5a). We placed two units, A and B, at each vertex of the lattice and summed their activity to obtain a local population activity as in the columnar recordings. Connections between units of any type are random. As expected, the autocorrelation of local population activity exhibits two distinct timescales corresponding to the self-excitation timescales of the two unit types (Fig. 5d).

The second model assumes that two timescales arise from two local biophysical processes, e.g., a fast membrane time constant and a slow synaptic time constant (Fig. 5b)[46]. We modeled the membrane time constant with the fast self-excitation timescale, and the synaptic time constant as a low-pass filter of the input to each unit with a slow time-constant $\tau_{synapse}$ (Methods)[46]. The connectivity between units is random. The autocorrelation of individual unit's activity in this model exhibit two timescales corresponding to the membrane ($\tau_{self}$) and synaptic ($\tau_{synapse}$) time constants (Fig. 5e).

Finally, in the third model, multiple timescales arise from recurrent dynamics shaped by the spatial network connectivity, akin to the horizontal connectivity in primate visual cortex[49]. Each model unit is connected to 8 nearby units (Fig. 5c). Although each unit has only a

single self-excitation timescale, the unit's autocorrelation exhibit multiple timescales with a fast decay at short time-lags and a slower decay at longer time-lags (Fig. 5f). The fast initial decay corresponds to the self-excitation timescale. The slow autocorrelation decay is generated by recurrent interactions among units in the network. In simulations, the slow autocorrelation decay closely matches the autocorrelation of the net recurrent input received by a unit from its neighbors (excluding the self-excitation input).

To understand how recurrent interactions generate slow timescales, we analytically computed the autocorrelation timescales of the unit's activity in the network with spatial connectivity, using the master equation for binary units with Glauber dynamics[50] (Methods, Supplementary Note 4, details in[47]). We found that the slow decay of the autocorrelation contains a mixture of interaction timescales $\tau_{int,k}$. Each $\tau_{int,k}$ arises from recurrent interactions on a different spatial scale, characterized by the modes of correlated fluctuations with different spatial frequencies **k** in the Fourier space (Methods). For each spatial frequency **k**, the interaction timescale depends on both the probability of horizontal interactions ($p_r$) and the self-excitation probability ($p_s$) (Methods, Eq. (24)). Shorter interaction timescales arise from higher spatial frequency modes (larger **k**) which correspond to persistent activity in local neighborhoods, and longer timescales are generated by more global interactions (smaller **k**)[47]. The longest timescale in the network is characterized by the global interaction timescale related to the zero spatial frequency mode (Methods, Eq. (25)). We can approximate the slow decay of the autocorrelation with a single effective interaction timescale ($\tau_{int}$) defined as a weighted average of all interaction timescales (Methods, Eq. (27)). Therefore, the autocorrelation shape is well approximated with two timescales: the fast self-excitation timescale and the slow effective interaction timescale.

Generating multiple timescales in spatial networks does not require strictly structured connectivity. Systematically changing the connectivity from structured to random reveals that networks with an intermediate level of local connectivity also exhibit multiple timescales in local dynamics (Fig. 6, Supplementary Note 5). However, by getting closer to a random connectivity, most interaction timescales become smaller and close to the self-excitation timescale, and only the global timescale does not depend on the network structure. Hence networks with different connectivity have the same global timescale (Fig. 6, inset). In fully random networks, the autocorrelation of a unit's activity effectively exhibits only two distinct timescales: the self-excitation timescale and the global interaction timescale. However, the global timescale has a very small relative contribution in local autocorrelations (scaled with the inverse number of neurons in the network) and is

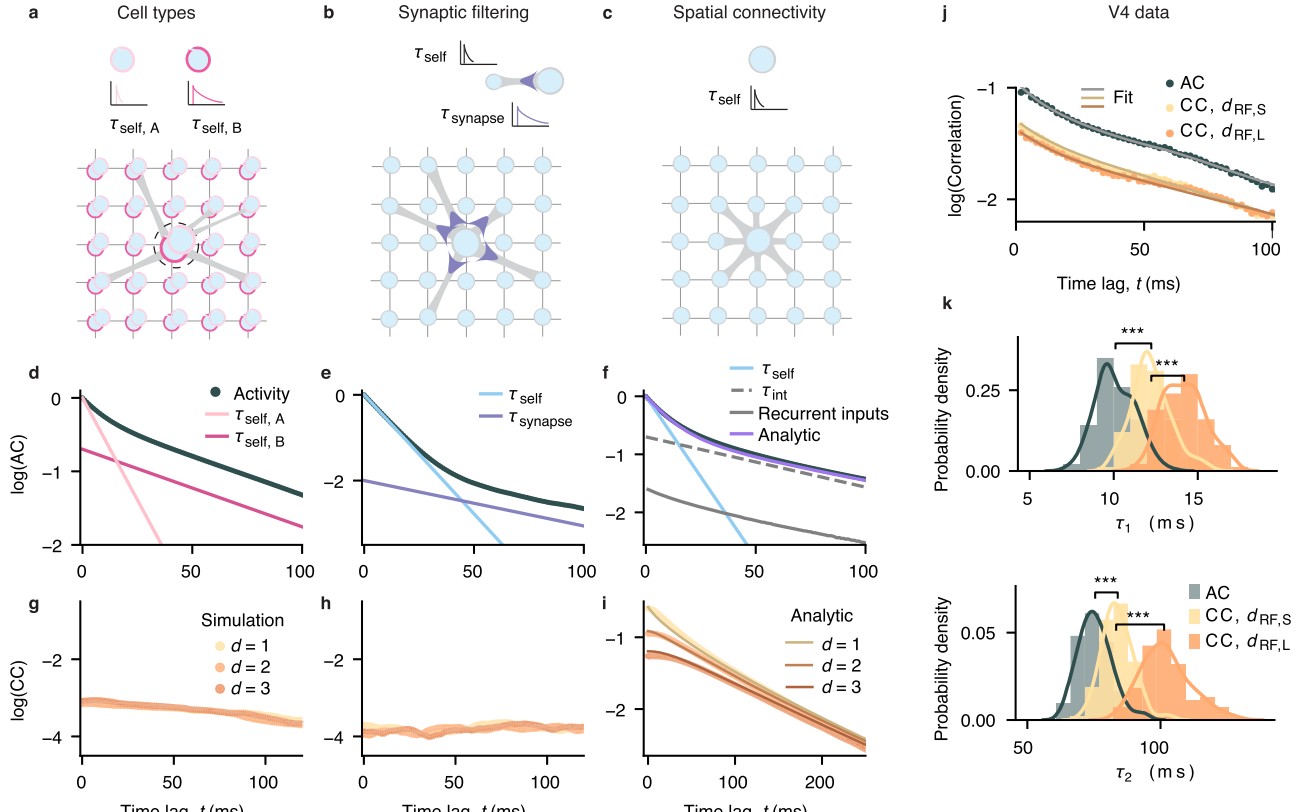

**Fig. 5 | Mechanisms for generating multiple timescales in local population activity. a–c** Network models consist of units (circles) arranged on a two-dimensional lattice (thin grey lines). Each target unit (large circle) receives inputs from 8 other units in the network (thick grey lines). The connectivity is random (**a**, **b**) or spatially arranged with each unit connected to its nearest neighbors (**c**). In the model with heterogeneous cell types (**a**), a local population at each lattice node (dashed circle) consists of two cell types, A and B, with distinct timescales (self-excitation probabilities $p_{s,A} = 0.88$ and $p_{s,B} = 0.976$). In the model with two local biophysical processes (**b**), each local population has a fast membrane time constant (modeled as $p_s = 0.88$) and a slow synaptic time constant (modeled as $\tau_{synapse} = 41$ ms). The spatial network model (**c**) assumes only a single self-excitation timescale ($p_s = 0.88$) for each unit. **d–f** All models reproduce two distinct timescales in the autocorrelations of local population activity. In the model with two cell types (**d**), the timescales correspond to the self-excitation timescales of two unit types ($\tau_{self,A}$, $\tau_{self,B}$, pink lines). In the model with synaptic filtering (**e**), the timescales correspond to the self-excitation and synaptic timescales ($\tau_{self}$, $\tau_{synapse}$, blue lines). In the spatial network model (**f**), the unit's autocorrelation exhibits multiple timescales and is well captured by the analytical derivation (purple). The fast autocorrelation decay corresponds to the self-excitation timescale ($\tau_{self}$, blue). The slower decay is

captured by the autocorrelation of recurrent inputs received by each unit in simulations (gray) and an analytical effective interaction timescale ($\tau_{int}$, dashed line). **g–i** In the models with random connectivity, cross-correlations between the activity of local populations do not depend on the distance ($d$) between units on the lattice (two cell types in **g**; synaptic filtering in **h**). In contrast, in the spatial network model, cross-correlations depend on distance $d$ and exhibit multiple timescales (**i**). The strength of cross-correlations decreases with distance, and slower interaction timescales dominate cross-correlations at longer distances. For all models: BP = 0.99, $p_{ext} = 10^{-4}$. **j** Auto- and cross-correlations of V4 spiking activity recorded on different channels overlaid with correlations of synthetic data with MAP parameters (data from monkey G, FT; data from monkey N in Supplementary Fig. 9). The strength of cross-correlations is smaller than the auto-correlation and decreases with RF-center distance ($d_{RF,L} > d_{RF,S}$). **k** Posterior distributions of timescales from fitting correlations in **j**. Cross-correlations had slower timescales than the autocorrelation, and slower timescales dominated cross-correlations at larger RF-center distances. Statistics: two-sided Wilcoxon rank-sum test, *** indicate $p < 10^{-10}$. Number of samples in each posterior $N = 100$. Correlations are plotted from the first time-lag ($t = 2$ ms). Source data are provided as a Source Data file.

hard to observe empirically as it requires data with excessively long trial duration.

While all three mechanisms account for multiple timescales in V4 autocorrelations, they can be distinguished in cross-correlations between local population activity at different spatial distances. In models with random connectivity, cross-correlations do not depend on distance between units on the lattice (Fig. 5g, h). In contrast, the model with spatial connectivity predicts that both the strength and timescales of cross-correlations depend on distance (Fig. 5i). Specifically, the zero time-lag cross-correlations decrease with distance. Moreover, cross-correlations contain multiple timescales equal to the interaction timescales in autocorrelations (Methods), but no self-excitation timescale since self-excitation is independent across units. With increasing distance, the weights of timescales generated by local interactions (high spatial frequency modes) decrease, and timescales generated by more global interactions (low spatial

frequency modes) dominate cross-correlations. Thus, cross-correlations become weaker and dominated by slower timescales at longer distances (analytical derivations in Methods, details in ref. 47). Approximating the shape of auto- and cross-correlations with two effective timescales, the theory predicts that both timescales in cross-correlations are larger than in the autocorrelation and increase with distance. Therefore, by measuring timescales of cross-correlations at different distances, we can determine which mechanism, spatial network interactions or local biophysical properties, is more consistent with neural dynamics in V4.

To test these model predictions in our V4 recordings, we computed cross-correlations between population activity on different channels during spontaneous activity (monkey G in FT, monkey N in AT2), which had the longest trial durations for better detection of slow timescales (Methods). Columnar recordings generally exhibit slight horizontal displacements which manifest in a systematic shift

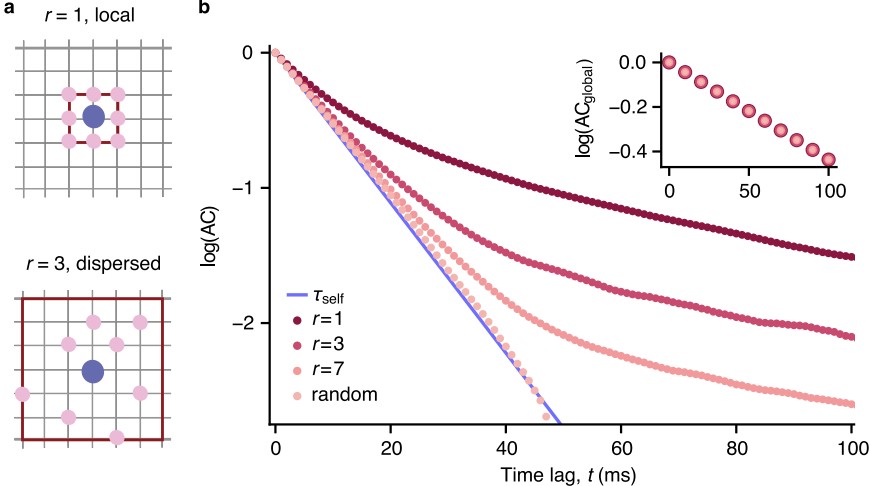

**Fig. 6 | Dependence of local but not global timescales on the spatial network structure. a** Schematic of local ($r = 1$) and dispersed ($r > 1$) spatial connectivity in the network model. Each unit (blue) is connected to 8 other units (pink) selected randomly within the connectivity radius $r$ (brown line). **b** Shapes of the auto-correlations of individual units (AC) reflect the underlying local connectivity structure. Interaction timescales disappear and the self-excitation timescale ($\tau_{self}$) dominates local autocorrelations when the connectivity radius increases while the connection strengths are kept constant ($p_s = 0.88$, $8p_r = 0.11$, $p_{ext} = 10^{-4}$). The autocorrelation of the global network activity ($AC_{global}$, inset) does not depend on the connectivity structure. Source data are provided as a Source Data file.

of receptive fields (RFs) across channels[51]. We used distances between the RF centers (RF-center distance) as a proxy for horizontal cortical distances[51]. For each monkey, we divided the cross-correlations into two groups with larger ($d_{RF,L}$) and smaller ($d_{RF,S}$) RF-center distances than the median distance (monkey G: $0 < d_{RF,S} < 2.08$, $2.08 < d_{RF,L} < 5$, monkey N: $0 < d_{RF,S} < 0.77$, $0.77 < d_{RF,L} < 2.25$, all distances are in degrees of visual angle, dva) and averaged the cross-correlations within each group. For comparison, we also computed the average auto-correlation of population activity on individual channels (i.e. without pooling spikes across channels). The differences between auto- and cross-correlations of V4 data appeared smaller than in the model since horizontal displacements between channels were relatively small, sampling mainly within the same or nearby columns[51].

The cross-correlations of V4 activity exhibited distinct fast and slow decay rates as predicted by the spatial network model (Fig. 5j, Supplementary Fig. 9, left). In agreement with the spatial network model, zero time-lag cross-correlations decreased with increasing RF-center distance (monkey G: mean for $d_{RF,S} = 0.047$, $d_{RF,L} = 0.040$, $p = 4 \times 10^{-4}$, $N = 152$; monkey N: mean for $d_{RF,S} = 0.022$, $d_{RF,L} = 0.013$, $p = 0.001$, $N = 128$, two-sided Wilcoxon rank-sum test), consistent with the reduction of pairwise noise correlations with lateral distance in V4[51,52]. The shapes of V4 auto- and cross-correlations were well approximated by fitted two-timescale generative models (Fig. 5j, Supplementary Fig. 9, left), and the estimated posterior distributions allowed us to compare auto- and cross-correlation timescales at different distances (Fig. 5k, Supplementary Fig. 9, right). Both fast and slow timescales were smaller in autocorrelations than in cross-correlations (Fast timescale: monkey G, mean $\tau_{1,AC} = 10.11$ ms, $\tau_{1,CC,S} = 12.24$ ms, $\tau_{1,CC,L} = 14.19$ ms; monkey N, mean $\tau_{1,AC} = 4.93$ ms, $\tau_{1,CC,S} = 12.18$ ms, $\tau_{1,CC,L} = 12.34$ ms; Slow timescale: monkey G, mean $\tau_{2,AC} = 75.46$ ms, $\tau_{2,CC,S} = 83.94$ ms, $\tau_{2,CC,L} = 101.94$ ms; monkey N, mean $\tau_{2,AC} = 26.53$ ms, $\tau_{2,CC,S} = 358.07$ ms, $\tau_{1,CC,L} = 552.70$ ms; number of samples in each posterior $N = 100$, all $p$-values $< 10^{-10}$, two-sided Wilcoxon rank-sum test). Both fast and slow timescales of cross-correlations increased with the RF-center distance in both monkeys, but the increase in the fast timescale did not reach statistical significance in monkey N ($\tau_2$: $p < 10^{-10}$, $\tau_1$: $p_G < 10^{-10}$, $p_N = 0.36$, two-sided Wilcoxon rank-sum test), possibly due to narrower range of RF-center distances in monkey N compared to monkey G (median $d_{RF,N} = 0.77$,

$d_{RF,G} = 2.08$ dva). Thus, predictions of the spatial network model, but not the models with random connectivity, were borne out by the data.

These results suggest that multiple timescales in local population activity in V4 arise from the recurrent dynamics shaped by the spatial connectivity of the primate visual cortex and not from local biophysical processes alone. Local biophysical mechanisms can also contribute to generating multiple neural timescales. For example, spatial connectivity combined with synaptic filtering can give rise to multiple autocorrelation timescales (Supplementary Fig. 10). The dependence of cross-correlation timescales on distance indicates that dominant timescales in the local population activity reflect the spatial network structure.

## Changes in the efficacy of network interactions modulate local timescales

We used the spatial network model to investigate which mechanisms can underlie the modulation of the slow timescales during attention. We matched the timescales between the model with local connectivity ($r = 1$) and experimental data to determine which changes in the model parameters can explain the attentional modulation of timescales in V4. We matched the self-excitation and effective interaction timescales of a model unit to, respectively, the fast and slow timescales of V4 activity (mean timescale ± s.e.m., Methods) for both the attend-away and attend-in (averaged over covert and overt) conditions (Fig. 7). We used a combination of analytical approximations and model simulations to find parameters that produce timescales similar to the V4 data (Methods).

We found that to reproduce the timescales in V4, the model needs to operate close to the critical point BP = 1 (Fig. 7b). At the critical point, each unit activates one other unit on average resulting in self-sustained activity[53]. Close to this regime, the timescales are flexible, such that small changes in the network excitability give rise to significant changes in timescales. To increase the slow timescale during attention, the total excitability of the network interactions should increase, shifting the network dynamics closer to the critical point. The overall increase in the interaction strength can be achieved by increasing the strength of either the self-excitation ($p_s$) or the recurrent interactions ($p_r$). Increasing $p_r$ while keeping $p_s$ constant allows for substantial changes in the slow timescale and a nearly unchanged fast timescale consistent with the V4 data. The increase of $p_s$ in the model

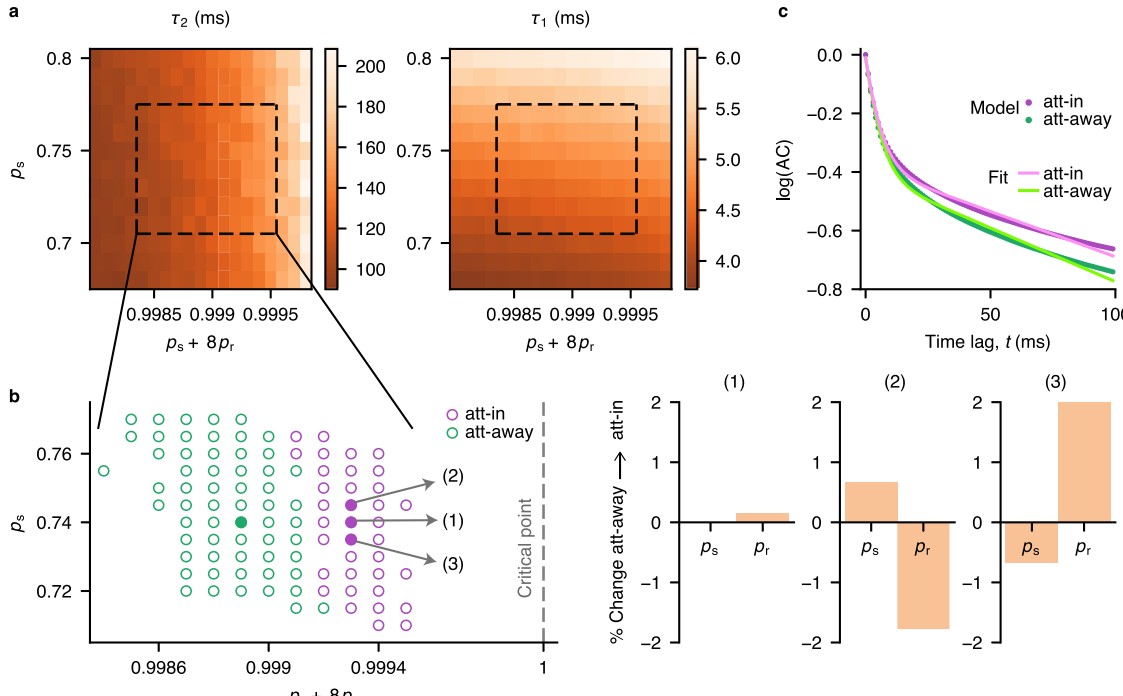

**Fig. 7 | Modulation of the slow timescale during attention is mediated by an increase in the efficacy of network interactions. a** Effect of connectivity parameters on local timescales in the model. The fast timescale ($\tau_1$, right) mainly depends on the self-excitation probability ($p_s$), whereas the slow timescale ($\tau_2$, left) depends on both the self-excitation ($p_s$) and recurrent horizontal interactions ($p_r$). The dashed rectangles indicate the range of parameters reproducing V4 timescales (mean ± s.e.m. of MAP estimates, Methods). **b** The slow timescale increases with the network excitability ($p_s + 8p_r$, left panel). Green and magenta dots indicate the parameters reproducing attend-away (att-away) and attend-in (att-in) timescales, respectively.

Filled dots show examples of experimentally observed 20% increase in $\tau_2$ for three possible scenarios based on different changes in $p_s$ or $p_r$ (right panels). Larger changes of parameters in scenarios (2) and (3) are due to coarser grid of $p_s$ used to fit the timescales. A similar change of $\tau_2$ can also be achieved with smaller changes in $p_s$ and $p_r$ (e.g., for all $0.74 < p_s < 0.745$ in scenario 2). **c** Example autocorrelations (ACs) from the model simulations with the attend-in and attend-away parameters for the scenario (2) in **b**. We fitted unbiased autocorrelations from the model simulations with double exponential functions (green and pink lines) to estimate the two timescales (Methods). Source data are provided as a Source Data file.

produces a slight increase in the fast timescale ($\tau_1$) (~ 0.4 ms on average), but such small changes in $\tau_1$ would be undetectable with our available data amount (the uncertainty of $\tau_1$ MAP estimate is ± 0.9 ms on average, Fig. 3b, e). The increase in $p_s$ can also be counterbalanced by a reduction in $p_r$ to produce the observed changes of timescales.

Several mechanisms can account for changes in the strength of recurrent interactions during attention. For example, the increase in $p_s$ is consistent with the observation that interactions between cortical layers in V4 increase during attention[42], when $p_s$ is interpreted as the strength of vertical recurrent interactions within cortical minicolumns. A reduction in $p_r$ can be mediated by neuromodulatory effects that reduce the efficacy of lateral connections in the cortex during attention[54]. In addition, our analytical derivations show that in the model with non-linear recurrent interactions, the effective strengths of recurrent interactions can also change by external input (Methods, details in ref. 47). The input alters the operating regime of network dynamics changing the effective strength of recurrent interactions. Thus, with non-linear interactions, timescales can be modulated by the input to the network, such as top-down inputs from higher cortical areas during attention[51,55]. Altogether, our model suggests that attentional modulation of timescales can arise from changes in the efficacy of recurrent interactions in visual cortex that can be mediated by neuromodulation or top-down attentional inputs.

## Discussion

We found that ongoing spiking activity of local neural populations within columns of the area V4 unfolded across fast and slow timescales, both in the presence and absence of visual stimuli. The slow timescale increased when monkeys attended to the receptive fields

location, showing that local intrinsic timescales can change flexibly from trial to trial according to selective attention. Furthermore, the slow timescales of neurons with RFs in the attended location correlated with the monkeys' reaction times suggesting that the increase in the slow timescale may contribute to behavioral benefits of selective spatial attention. To understand the mechanisms underlying the multiplicity and flexible modulation of timescales, we developed network models linking intrinsic timescales to biophysical properties of individual neurons or the spatial connectivity structure of the visual cortex. Only the spatial network model correctly predicted the distance-dependence of spatiotemporal correlations in V4, indicating that multiple timescales in V4 dynamics arise from the spatial connectivity of primate visual cortex. The model suggests that slow timescales increase with the effective strength of recurrent interactions.

### Multiple intrinsic timescales in neural activity

Previous studies characterized the autocorrelation of ongoing neural activity with a single intrinsic timescale[11,13,15,16]. The intrinsic timescale was usually measured for neural populations either by averaging autocorrelations of single neurons in one area[11] or using coarse-grained measurements such as ECoG[13] or fMRI[15,16]. Thus, ongoing dynamics in each area were described with a single intrinsic timescale that varied across areas. We extended this view by showing that, within one area, local population activity exhibits multiple intrinsic timescales. These timescales reflect ongoing dynamics on single trials and are not driven by task events. Our results suggest that the multiplicity of timescales is an intrinsic property of neural activity arising from inherent cellular and network properties of the cortex.

We show that multiple timescales in local dynamics can emerge from the spatial connectivity structure in a recurrent network model. The presence of two dominant timescales ($\tau_{\text{self}}$, $\tau_{\text{int}}$) in local dynamics depends on the combination of the structured connectivity and strong, mean-driven interactions between units. Networks with random connectivity (Fig. 6, b) or weak, diffusion-type interactions[51] exhibit only one dominant timescale in local activity (Supplementary Note 6). Moreover, local biophysical properties alone cannot explain the dependence of spatiotemporal neural correlations on lateral distance in the cortex, highlighting the importance of spatial network interactions for generating multiple timescales in local population activity.

In our network model with local spatial connectivity, recurrent interactions across different spatial scales induce multiple slow timescales. To generate multiple slow timescales, our network operates close to a critical point. Spiking networks with spatial connectivity can generate fast correlated fluctuations that emerge from instability at particular spatial frequency modes[56]. Slow fluctuations of firing rates can also arise in networks with clustered random connectivity, but interactions between clusters induce only a single slow timescale[34]. We show that more local spatial connectivity (smaller $r$) leads to slower dynamics and modifies the weights and composition of timescales in the local activity. The timescale of the global activity, on the other hand, is the same across networks with distinct local timescales and different connectivity structures. These results show that local temporal and spatial correlations of neural dynamics are closely tied together.

In our model, integrating activity over larger spatial scales leads to disappearance of faster interaction timescales (higher spatial frequencies) leaving only slower interaction timescales (lower spatial frequencies) in the coarse-grained activity. At the extreme, the global network activity exhibits only the slowest interaction timescale (the global timescale). This mechanism may explain the prominence of slow dynamics in meso- and macroscale measures of neural activity such as LFP or fMRI[57], while faster dynamics dominate in local measures such as spiking activity. The model predicts that the slowest interaction timescales have very small weights in the autocorrelation of local neural activity and thus can be detected in local activity only with excessively long recordings. Indeed, infraslow timescales (on the order of tens of seconds and minutes) are evident in the cortical spiking activity recorded over hours[58].

## Functional relevance of neural activity timescales
Intrinsic timescales are thought to define the predominant role of neurons in the cognitive processes[17]. For example, in the orbitofrontal cortex, neurons with long intrinsic timescales are more involved in decision-making and the maintenance of value information[44]. In the prefrontal cortex (PFC), neurons with short intrinsic timescales are primarily involved in the early phases of working memory encoding[31], while neurons with long timescales play a significant role in coding and maintaining information during the delay period[31,45]. Our finding that intrinsic timescales can flexibly change from trial to trial (and across epochs within a trail[13]) suggests a possibility that task-induced timescales may correspond with intrinsic timescales only during specific task phases. These results may explain why the task-induced timescales of single neurons do not correlate with intrinsic timescales measured over the entire task duration[12].

We found that timescales of local neural activity changed from trial to trial depending on the attended location. A previous ECoG study found that the intrinsic timescale of neural activity in cortical association areas increased after engagement in a working memory task[13]. Our findings go beyond this earlier work by showing that the modulation of timescales can be functionally specific as it selectively affects only neurons representing the attended location within the retinotopic map. While changes in timescale due to task engagement could be mediated by slow global processes such as arousal, the retinotopically precise modulation of timescales requires local changes targeted to task-relevant neurons. Our results further show that the modulation of timescales also occurs in sensory cortical areas and cognitive processes other than memory maintenance[13] which explicitly requires temporal integration of information. The correlation of slow timescales with reaction times during attention may be functionally relevant, potentially allowing neurons to integrate information over longer durations.

Longer timescales during attention in the model are associated with shifting the network dynamics closer to a critical point. Shifting closer to criticality was also suggested as a mechanism for the increase in gamma-band synchrony and stimulus discriminability during attention[59]. Furthermore, strong recurrent dynamics close to the critical point can flexibly control the dimensionality of neural activity[60]. Hence, operating closer to the critical point during attention might help to optimize neural responses to environmental cues and improve information processing[61].

## Mechanisms for attentional modulation of timescales
Changes in the slow timescale of neural activity due to attention occurred from one trial to another. Such swift changes cannot be due to significant changes in the underlying network structure and require a fast mechanism. Our model suggests that the modulation of slow timescales during attention can be explained with a slight increase in the network excitability mediated by an increase in the efficacy of horizontal recurrent interactions, or by an increase in the efficacy of vertical interactions accompanied by a decrease in the strength of horizontal interactions.

Several physiological processes may underlie these network mechanisms in the neocortex. Top-down inputs during attention can enhance the local excitability in cortical networks[55]. Our analytical derivations show that inputs can increase the effective strength of recurrent interactions between neurons in networks with non-linear interactions, similar to previous models[18,62]. Similar modulation of timescales during covert and overt attention suggests that top-down attentional inputs arrive from brain areas that represent both attention-related and saccade-related information. Frontal eye field (FEF) can be a possible source for such modulations[37,63,64]. Furthermore, feedback connections from higher visual areas like PFC or the temporal-occipital area (TEO) to lower visual areas have broader terminal arborizations than the size of the receptive fields in lower areas[65,66]. These feedback inputs can coordinate activity across minicolumns in V4. Moreover, vertical interactions in V4 measured with local field potentials (LFPs) increase during attention[42], while neuromodulatory mechanisms can reduce horizontal interactions. The level of Acetylcholine (ACh) can modify the efficacy of synaptic interactions during attention in a selective manner[54]. Increase in ACh strengthens the thalamocortical synaptic efficacy by affecting nicotinic receptors and reduces the efficacy of horizontal recurrent interactions by affecting muscarinic receptors. Decrease in horizontal interactions is also consistent with the proposed reduction of spatial correlations length during attention[51]. These observations suggest that an increase in vertical interactions and a decrease in horizontal interactions is a likely mechanism for modulation of the slow timescale during attention.

To identify biophysical mechanisms of timescales modulation, experiments with larger number of longer trials are required to provide tighter bounds for estimated timescales. Additionally, detailed biophysical models can help distinguish different mechanisms, since biophysical and cell-type specific properties of neurons might also be involved in defining neural timescales[67,68]. In particular, diverse timescales observed across single neurons within one area[17,31,44,45,69] require models considering a heterogeneous parameter space and can have computational implications for the brain[70]. Here, we used the RF-center distances as a proxy for spatial distances in the cortex. Experiments with spatially organized recording sites would allow to study the relation between temporal and spatial correlations more directly. Furthermore, developing recurrent network models that perform the

selective attention task can help to find direct links between the modulation of dynamics and task performance. Finally, perturbation experiments that modulate selectively top-down inputs or neuromodulatory levels can provide the most direct test of the underlying mechanisms.

Our findings reveal that targeted neural populations can integrate information over variable timescales following changes in the cognitive state. Our model suggests that local interactions between neurons via the spatial connectivity of primate visual cortex can underlie the multiplicity and flexible modulation of intrinsic timescales. Our experimental observations combined with the computational model provide a basis for studying the link between the network structure, functional brain dynamics, and flexible behavior.

## Methods

### Behavioral tasks and electrophysiology recordings
We used previously published datasets[36,37,71–73]. Experimental procedures for the fixation task and attention task 1 were in accordance with NIH Guide for the Care and Use of Laboratory Animals, the Society for Neuroscience Guidelines and Policies, and Stanford University Animal Care and Use Committee. Experimental procedures for the attention task 2 were in accordance with the European Communities Council Directive RL 2010/63/EC, and Use of Animals for Experimental Procedures, and the UK Animals Scientific Procedures Act. Three male monkeys (*Macaca mulatta*, between 6 to 9 years old) were used in the experiments. Monkeys were motivated by scheduled fluid intake during experimental sessions and juice reward.

On each trial of the fixation task (FT, monkey G), the monkey was rewarded for fixating a central dot on a blank screen for 3 s. In attention task 1 (AT1, monkeys G, B), the monkey detected orientation changes in one of the four peripheral grating stimuli while maintaining central fixation. Each trial started by fixating a central fixation dot on the screen and after several hundred milliseconds (170 ms for monkey B and 333 ms for monkey G), four peripheral stimuli appeared. Following a 200 – 500 ms period, a central attention cue indicated the stimulus that was likely to change with ~ 90% validity. Cue was a short line from fixation dot pointing toward one of the four stimuli, randomly chosen on each trial with equal probability. After a variable interval (600 – 2200 ms), all four stimuli disappeared for a brief moment and reappeared. Monkeys were rewarded for correctly reporting the change in orientation of one of the stimuli (50% of trails) with an antisaccade to the location opposite to the change, or maintaining fixation if none of the orientations changed. Due to the anticipation of antisaccade response, the cued stimulus was the target of covert attention, while the stimulus in location opposite to the cue was the target of overt attention. In attend-in conditions, the cue pointed either to the stimulus in the RFs of the recorded neurons (covert attention) or to the stimulus opposite to the RFs (overt attention). The remaining two cue directions were attend-way conditions.

In attention task 2 (AT2, Newcastle cohort monkey, N), the monkey detected a small luminance change within the white phase of a square wave static grating. The monkey initiated a trial by holding a bar and visually fixating a fixation point. The color of the fixation point indicated the level of spatial certainty (red: narrow focus, blue: wide focus). After 500 ms a cue appeared indicating the location and focus of the visual field to attend to. The cue was switched off after 250 ms. After another second two gratings appeared, one in the center of the RFs and one diametrically opposite with respect to the fixation point. The grating at the position indicated by the cue was the test stimulus. The other grating served as the distractor. After at least 500 ms a small luminance change (dimming) occurred either in the center of the grating (narrow focus) or in one of 12 peripheral positions (wide focus). If the dimming occurred in the distractor grating first, the monkey had to ignore it. The monkey was rewarded for a bar release within 750 ms

of the dimming in the test grating. The faster the monkey reacted, the larger reward it received. Two grating sizes (small and large) were used in this experiment. We analyzed trials with the small grating to avoid surround-suppression effects created by the large grating sizes extending beyond the neurons' summation area[74].

Recordings were performed in the visual area V4 with linear array microelectrodes inserted perpendicularly to the cortical layers. Data were amplified and recorded using the Omniplex system (Plexon) in AT1 and FT and with the Digital Lynx recording system (Neuralynx) in AT2. Arrays were placed such that receptive fields of recorded neurons largely overlapped. Each array had 16 channels with 150 μm center-to-center spacing. In AT1 and FT, all 16 channels were visually responsive. In AT2, the number of visually-responsive channels per recording ranged between 8 and 12 with the median at 9.

### Computing autocorrelations of neural activity
We computed autocorrelations from multi-unit (MUA) spiking activity recorded in the presence (stimulus-driven) and absence (spontaneous) of visual stimuli (brown and yellow frames in Supplementary Fig. 1). For spontaneous activity, we analyzed spikes during the 3s fixation epoch in FT, and during the 800 ms epoch from 200 ms after the cue offset until the stimulus onset in AT2. For stimulus-driven activity, we analyzed spikes in the epoch from 400 ms after the cue onset until the stimulus offset in AT1, and from 200 ms after the stimulus onset until the dimming in AT2. For the stimulus-driven activity, trials in both attention tasks had variable durations (500 – 2200 ms). Thus, we computed autocorrelations in non-overlapping windows of 700 ms for AT1 and 500 ms for AT2. On long trials, we used as many windows as would fit within the trial duration, and we discarded trials that were shorter than the window size. The duration of windows were selected such that we had at least 50 windows for each condition in each session. 3 out of 25 recording sessions in monkey G (AT1) were excluded due to short trial durations. For spontaneous activity, the windows were 3 s in FT and 800 ms in AT2.

We computed the average spike-count autocorrelation for each recording session. On each trial we pooled the spikes from all visually-responsive channels and counted the pooled spikes in 2 ms bins. For each behavioral condition (stimulus orientation, attention condition), we averaged spike-counts at each time-bin across trials, and subtracted the trial-average from the spike-counts at each bin[11] to remove correlations due to changes in firing rate locked to the task events. We segmented the mean-subtracted spike-counts $A(t'_i)$ into windows of the same length $N$, where $t'_i$ ($i = 1...N$) indexes bins within a window. We then computed the autocorrelation in each window as a function of time-lag $t_j$[39]:

$$\text{AC}(t_j) = \frac{1}{\hat{\sigma}^2(N-j)}\sum_{i=1}^{N-j}\left(A(t'_i) - \hat{\mu}_1(j)\right)\left(A(t'_{i+j}) - \hat{\mu}_2(j)\right). \quad (1)$$

Here $\hat{\sigma}^2 = \frac{1}{N-1}\sum_{i=1}^{N}(A(t'_i)^2 - \frac{1}{N^2}(\sum_{i=1}^{N}A(t'_i))^2)$ is the sample variance, and $\hat{\mu}_1(j) = \frac{1}{N-j}\sum_{i=1}^{N-j}A(t'_i)$ and $\hat{\mu}_2(j) = \frac{1}{N-j}\sum_{i=j+1}^{N}A(t'_i)$ are two different sample means. In Eq. (1) for autocorrelation, we subtracted window-specific mean to remove correlations due to slow changes in firing rate across trials, such as slow fluctuations related to changes in the arousal state. Thus, the range of timescales was limited to the trial duration. These timescales reflect the intrinsic neural dynamics within single trials. Finally, we averaged the autocorrelations over windows of the same behavioral condition separately for each recording session. The exact method of computing autocorrelations does not affect the estimated timescales, since we use the same method for computing autocorrelations of synthetic data when fitting generative models with the aABC method[39].

In AT1, we averaged autocorrelations over trials with different stimulus orientation for each attention condition, since all attention conditions contained about the same number of trials with each orientation. For stimulus-driven activity in AT2, we first estimated

timescales separately for focus wide and narrow conditions and found no significant differences (two-sided Wilcoxon signed rank test between MAP estimates, $p > 0.05$). Thus, we averaged autocorrelations of the focus narrow and wide conditions and refitted the average autocorrelations. The same procedure was applied to the spontaneous activity in AT2, and since there was no significant difference in timescales between different focus or attention conditions (two-sided Wilcoxon signed rank test between MAP estimates for the two-by-two conditions, $p > 0.05$), we averaged the autocorrelations over all conditions and refitted the average autocorrelation.

For estimating the timescales, we excluded sessions with autocorrelations dominated by noise or strong oscillations that could not be well described with a mixture of exponential decay functions. We excluded a session if the autocorrelation fell below 0.01 (log(AC) fell below −2) in lags smaller or equal to 20 ms (Supplementary Fig. 11). Based on this criterion, we excluded 3 out of 22 sessions for monkey G in AT1, 8 out of 21 sessions during covert attention and 9 out of 21 during overt attention for monkey B in AT1, 2 out 20 sessions for spontaneous activity and 8 out 20 sessions for stimulus-driven activity for Monkey N in AT2. The difference in the number of excluded sessions for Monkey N during spontaneous and stimulus-driven activity is explained by the larger amount of data available for computing autocorrelations during spontaneous activity due to averaging over attention conditions and longer window durations (800 ms vs. 500 ms).

For visualization of autocorrelations, we omitted the zero time-lag ($t = 0$ ms) (examples with the zero time-lag are shown in Supplementary Fig. 11). The autocorrelation drop between the zero and first time-lag ($t = 2$ ms) reflects the difference between the total variance of spike counts and the variance of instantaneous rate according to the law of total variance for a doubly stochastic process[39]. This drop is fitted by the aABC algorithm when estimating the timescales.

## Statistics and reproducibility

Three male monkeys were used in the experiments which is a standard sample size for primate studies[18,37,42,43]. Animals' ability to perform the task determined the number of trials in each recording session. The number of simultaneously recorded neurons was determined by the properties of the linear multielectrode arrays used for the experiments. Blinding of the investigators were not relevant, since there were no differences between the subjects that would conceivably create biases. Sex of the subjects was not considered in study design, as the sample size is too small to make any meaningful statements about the impact of sex on the mechanisms we found. Participants were not allocated into groups and no subject randomization was implemented. Information about excluded recording sessions is provided in the previous section.

## Estimating timescales with adaptive Approximate Bayesian Computations

We estimated the autocorrelation timescales using the aABC method that overcomes the statistical bias in empirical autocorrelations and provides the posterior distributions of unbiased estimated timescales[39]. The width of inferred posteriors indicates the uncertainty of estimates. For more reliable estimates of timescales (i.e. narrower posteriors), we selected epochs of experiments with longer trial durations (brown and yellow frames in Supplementary Fig. 1).

The aABC method estimates timescales by fitting the spike-count autocorrelation with a generative model. We used a generative model based on a doubly stochastic process with one or two timescales. Spike counts were generated from a rate governed by a linear mixture of Ornstein-Uhlenbeck (OU) processes (one OU process $A_{\tau_k}$ for each timescale $\tau_k$)

$$A_{\text{OU}}(t') = \sum_{k=1}^{n} \sqrt{c_k} A_{\tau_k}(t'), \quad \sum_{k=1}^{n} c_k = 1, \quad c_k \in [0,1], \quad (2)$$

where $n$ is the number of timescales and $c_k$ are their weights. The aABC algorithm optimizes the model parameters to match the spike-count autocorrelations between V4 data and synthetic data generated from the model. We generated synthetic data with the same number of trials, trial duration, mean and variance of spike counts as in the experimental data. By matching these statistics, the empirical autocorrelations of the synthetic and experimental data are affected by the same statistical bias when their shapes match. Therefore, the timescales of the fitted generative model represent the unbiased estimate of timescales in the neural data.

The spike-counts $s$ are sampled for each time-bin $[t'_i, t'_{i+1}]$ from a distribution $p_{\text{count}}(s|\lambda(t'_i))$, where $\lambda(t'_i) = A_{\text{OU}}(t'_i)\Delta t'$ is the mean spike-count and $\Delta t' = t'_{i+1} - t'_i$ is the bin size. To capture the possible non-Poisson statistics of the recorded neurons, we introduce a dispersion parameter $\alpha$ defined as the variance over mean ratio of the spike-counts distribution $\alpha = \sigma^2_{s|\lambda(t'_i)}/\lambda(t'_i)$. For a Poisson distribution, $\alpha$ is equal to 1. We allow for non-Poisson statistics by sampling the spike counts from a gamma distribution and optimize the value of $\alpha$ together with the timescales and the weights.

On each iteration of the aABC algorithm, we draw sample parameters from a prior distribution (first iteration) or a proposal distribution (subsequent iterations) defined based on the prior distribution and parameters accepted on the previous iteration. Then, we generate synthetic data from the sampled parameters and compute the distance $d$ between the autocorrelations of synthetic and experimental data:

$$d(t_m) = \frac{1}{m} \sum_{j=0}^{m} \left( \text{AC}_{\text{experimental}}(t_j) - \text{AC}_{\text{synthetic}}(t_j) \right)^2, \quad (3)$$

where $t_m$ is the maximum time-lag considered in computing the distance. We set $t_m$ to 100 ms to avoid over-fitting the noise in the tail of the autocorrelations. If the distance is smaller than a predefined error threshold $\varepsilon$, the sample parameters are accepted and added to the posterior distribution. Each iteration continued until 100 sample parameters were accepted. The initial error threshold was set to $\varepsilon_0 = 0.1$, and in subsequent iterations, the error threshold was updated to the first quartile of the distances for the accepted samples. The fraction of accepted samples out of all drawn parameter samples is recorded as the acceptance rate accR. The algorithm stops when the acceptance rate reaches accR < 0.0007. The final accepted samples are considered as an approximation for the posterior distribution. We computed the MAP estimates by smoothing the final joint posterior distribution with a multivariate Gaussian kernel and finding its maximum with a grid search.

The choice of summary statistic (e.g., autocorrelations in the time domain or power spectra in the frequency domain and the fitting range) does not affect the accuracy of estimated timescales and only changes the width of the estimated posteriors[39]. The frequency-domain fitting converges faster in wall-clock time than time-domain fitting[39]. As a control, we also estimated timescales by fitting the whole shape of power spectral density in the frequency domain. The results of these fits (Supplementary Fig. 7) were in agreement with the time-domain fits with a limited fitting range (Fig. 3).

We used a multivariate uniform prior distribution over all parameters. For the two-timescale generative model ($M_2$), the priors' ranges were set to

$$\tau_1 : U[0, 60], \quad \tau_2 : U[0, 400], \quad c_1 : U[0, 1], \quad \alpha : U[0.7, 1.3], \quad (4)$$

and for the one-timescale generative model ($M_1$) they were set to

$$\tau : U[0, 400], \quad \alpha : U[0.7, 1.3]. \quad (5)$$

## Model comparison with adaptive Approximate Bayesian Computations

We used the inferred posteriors from the aABC fit to determine whether the V4 data autocorrelations were better described with the one-timescale ($M_1$) or the two-timescale ($M_2$) generative models[39]. First, we measured the goodness of fit for each model based on the distribution of distances between the autocorrelation of synthetic data from the generative model and the autocorrelation of V4 data. We approximated the distributions of distances by generating 1000 realizations of synthetic data from each model with parameters drawn from the posterior distributions and computing the distance for each realization. If the distributions of distances were significantly different (two-sided Wilcoxon ranksum test), we approximated the Bayes factor, otherwise the summary statistics were not sufficient to distinguish these two models[75].

Bayes factor is the ratio of marginal likelihoods of the two models and takes into account the number of parameters in each model[76]. In the aABC method, the ratio between the acceptance rates of two models for a given error threshold $\varepsilon$ approximates the Bayes factor (BF) for that error threshold[39]:

$$\mathrm{BF}(\varepsilon) = \frac{\mathrm{accR}_{M_2}(\varepsilon)}{\mathrm{accR}_{M_1}(\varepsilon)}. \tag{6}$$

Acceptance rates can be computed using the cumulative distribution function (CDF) of the distances for a given error threshold $\varepsilon$,

$$\mathrm{CDF}_{M_i}(\varepsilon) = p_{M_i}(d < \varepsilon) = \mathrm{accR}_{M_i}(\varepsilon), \quad i = 1, 2, \tag{7}$$

where $p_{M_i}(d)$ is the probability distribution of distances for model $M_i$. Thus, the ratio between the CDF of distances approximates the Bayes factor for every chosen error threshold. To eliminate the dependence on a specific error threshold, we computed the acceptance rates and the Bayes factor for varying error thresholds. Since only small errors indicate a well-fitted model, we computed the Bayes factor for all error thresholds that were smaller than the largest median of distance distributions of two models.

The $M_2$ model was selected if its distances were significantly smaller than for the $M_1$ model (two-sided Wilcoxon ranksum test) and $\mathrm{CDF}_{M_2}(\varepsilon) > \mathrm{CDF}_{M_1}(\varepsilon)$, i.e. BF > 1, for all $\varepsilon < \max_{M_1,M_2}[\mathrm{median}(\varepsilon)]$ (Supplementary Fig. 2). The same procedure was applied for selecting the $M_1$ model. Although the Bayes factor threshold was set at 1, in most cases we obtained BF $\gg$ 1, indicating strong evidence for the two-timescale model. If the distribution of distances for the two models were not significantly different or the condition for the ratio between CDFs did not hold for all selected $\varepsilon$ (CDFs were crossing), we classified the outcome as inconclusive, meaning that data statistics were not sufficient to make the comparison.

## Timescales of auto- and cross-correlations of spiking activity on individual channels

We computed the average auto- and cross-correlations of the multi-unit spiking activity recorded on individual channels during spontaneous activity (monkey G in FT, Monkey N in AT2). We computed the autocorrelation of each channel's activity using the same procedure described above and then averaged the autocorrelations across channels and recording sessions for each monkey. We computed the cross-correlations between spike counts on every pair of channels ($A_a$ and $A_b$) that were at least two channels apart ($|a - b| \geq 2$ e.g., channels 1 and 3) as a function of time-lag $t_j$

$$\mathrm{CC}_{a,b}(t_j) = \frac{1}{\sqrt{\hat{\sigma_a}^2 \hat{\sigma_b}^2}(N-j)} \sum_{i=1}^{N-j} \left( A_a(t_i') - \hat{\mu}_a(j) \right) \left( A_b(t_{i+j}') - \hat{\mu}_b(j) \right). \tag{8}$$

Here $\hat{\sigma_a}^2$ and $\hat{\sigma_b}^2$ are the sample variances, and $\hat{\mu}_a(j) = \frac{1}{N-j}\sum_{i=1}^{N-j} A_a(t_i')$ and $\hat{\mu}_b(j) = \frac{1}{N-j}\sum_{i=j+1}^{N} A_b(t_i')$ are the sample means for the activity on each channel. Then, we divided the cross-correlations for each monkey in two groups based on the monkey-specific median RF-center distance and averaged over the cross-correlations within each group.

The mapping of RFs was described previously[36]. RFs were measured by recording spiking responses to brief flashes of stimuli on an evenly spaced $6 \times 6$ grid covering the lower left visual field (FT) or an evenly spaced $12 \times 9$ grid centered on the RF (AT2). Spikes in the window $0 - 200$ ms (FT) or $50 - 130$ ms (AT2) relative to the stimulus onset were averaged across all presentations of each stimulus. First, we assessed the statistical significance of a given RF[77] and only included channels with a significant RF. Then, we found the RF center as the center of mass of the response map, and estimated the horizontal displacements between channels by computing the distances between their RF centers.

We estimated the timescales of auto- and cross-correlations using the aABC method. We assumed the correlation between channels' activity can be modeled as a two-timescale OU process shared between the two channels. We fitted the cross-correlation shape by the unnormalized autocorrelation of the shared OU process, such that the variance of the OU process (i.e. the autocorrelation at lag zero) defines the strength of correlations. Thus, we used a two-timescale OU process as the generative model and applied the aABC method to optimize the model parameters by minimizing the distance between the auto-correlation of synthetic data from the OU process and V4 cross-correlations. The aABC method returned a multivariate posterior distribution for timescales, their weights and the variance of the OU process. We computed the distances starting from the first time-lag $t = 2$ ms up to $t_m = 100$ ms. For a fair comparison between the auto- and cross-correlations timescales, we used the same procedure to estimate the timescales of individual channels' autocorrelations. For fitting the autocorrelation of monkey G, we additionally excluded the second time-lag $t = 2$ ms, since AC($t = 2$) < AC($t = 4$), potentially related to refractory period of neurons (similar to[11,31,44]).

## Testing correlation between timescales and reaction times with linear mixed-effects models

To compute the reaction times for each attention condition, we separated the trials between attend-in (separate covert and overt) and attend-away conditions. We computed the average reaction times of the monkeys for each recording session and each condition as the average duration between the reappearance of the stimuli and initiation of the anti-saccade response (AT1, only trials with a change in stimuli orientation) or the average duration between dimming in the target stimulus and the bar release (AT2), across trials with the same attention condition.

We quantified the relationship between average reaction times and MAP estimates of the fast and slow timescales in each session for two different attention conditions (attend-in and attend-away). For this analysis, we pulled the data across covert and overt attend-in conditions, resulting in more samples for the attend-in than attend-away condition. For each attention condition, we fitted a separate linear mixed-effects model using the "*fitlm*" function in the MATLAB R2021a. In these models, we consider data from each monkey as a separate group (i.e. a random effect) with a separate intercept to account for individual differences between the monkeys and between the two response types in the attention tasks (anti-saccade versus bar release).

We fitted two different models that considered either one or two fixed effects for each attention condition. First, we fitted models that considered as the fixed effect, either the slow timescale ($\tau_{2,\mathrm{cond}}$)

$$\mathrm{RT}_{i,m} = \omega_0 + \omega_1 \tau_{2,\mathrm{cond},i} + \Omega_{0,m} + \varepsilon_{i,m}, \tag{9}$$

or the fast timescale ($\tau_{1,\text{cond}}$),

$$\text{RT}_{i,m} = \omega_0 + \omega_1 \tau_{1,\text{cond},i} + \Omega_{0,m} + \varepsilon_{i,m}. \tag{10}$$

Here the index cond denotes attend-in or attend-away condition, RT indicates the reaction time, $i$ is the session index, and $m \in \{G, B, N\}$ indicates three different monkeys. $\omega_0$ and $\omega_1$ give the intercept and slope of the fixed effect with a given p-value. $\Omega_{0,m}$ and $\varepsilon_{i,m}$ are the random effects, where $\Omega_{0,m}$ gives a monkey-specific intercept and $\varepsilon_{i,m}$ gives the residuals. We also fitted models that considered both fast and slow timescales as fixed effects simultaneously,

$$\text{RT}_{i,m} = \omega_0 + \omega_1 \tau_{2,\text{cond},i} + \omega_2 \tau_{1,\text{cond},i} + \Omega_{0,m} + \varepsilon_{i,m}. \tag{11}$$

These models return two fixed-effect coefficients $\omega_{1,2}$ with p-values, one for each timescale. The resulting statistics for the two fitted models were consistent (Supplementary Table 1, 2). In the main text, we reported statistics from the first model type (Fig. 4, Supplementary Table 1).

## Network model with spatially structured connections
The network model operates on a two-dimensional square lattice of size $100 \times 100$ with periodic boundary conditions. Each unit in the model is connected to 8 other units taken either from its direct Moore neighborhood (local connectivity, Fig. 6a, top) or randomly selected within the connectivity radius $r$ (dispersed connectivity, Fig. 6a, bottom). Activity of each unit is represented by a binary state variable $S_i \in \{0, 1\}$ ($i = 1...N$, where $N = 10^4$ is the number of units). The units act as probabilistic integrate-and-fire units[78] following linear or non-linear integration rules. States of the units are updated in discrete time-steps $t'$ based on a self-excitation probability ($p_s$), probability of excitation by the connected units ($p_r$), and the probability of external excitation ($p_{\text{ext}} \ll 1$). The transition probabilities for each unit $S_i$ at time-step $t'$ are either governed by additive interaction rules (linear model):

$$p(S_i = 0 \rightarrow 1) = p_{\text{ext}} + p_r \sum_j S_j,$$
$$p(S_i = 1 \rightarrow 0) = 1 - \left(p_{\text{ext}} + p_s + p_r \sum_j S_j\right), \tag{12}$$

or multiplicative interaction rules (non-linear model):

$$p(S_i = 0 \rightarrow 1) = 1 - (1 - p_{\text{ext}})(1 - p_r)^{\sum_j S_j},$$
$$p(S_i = 1 \rightarrow 0) = (1 - p_{\text{ext}})(1 - p_s)(1 - p_r)^{\sum_j S_j}. \tag{13}$$

Here, $\sum_j S_j$ indicates the number of active neighbors of unit $S_i$ at time-step $t'$. For the analysis in the main text, we used the linear model. The non-linear model generates similar local temporal dynamics (Supplementary Fig. 12). In the linear model, the sum of connection probabilities $\text{BP} = p_s + 8p_r$ is the branching parameter that defines the state of the dynamics relative to a critical point at $\text{BP} = 1$[53,78].

To compute the average local autocorrelation in the network, we simulated the model for $10^5$ time-steps and averaged the autocorrelations of individual units. The global autocorrelations were computed from the pooled activity of all units in the network. To compute the autocorrelation of horizontal inputs for a unit $i$, we simulated the network with an additional "shadow" unit, which was activated by the same horizontal inputs ($p_r$) as the unit $i$ but without the inputs $p_s$ and $p_{\text{ext}}$. The shadow unit did not activate other units in the network. The autocorrelation of horizontal recurrent inputs was computed from the shadow unit activity. We computed the cross-correlations between the activity of each pair of units in the network and averaged the cross-correlations over pairs with the same distance $d$ between units. To have the same number of sample cross-correlations for each distance, we

randomly selected $4 \times 10^4$ pairs per distance. The spatial distance in the model is defined as the Chebyshev distance on the lattice (e.g., $d = 1$ is the Moore neighborhood). Each simulation started with a random configuration of active units based on the analytically computed steady-state mean activity (Eq. (21)). Running simulations for long periods allowed us to avoid the statistical bias in the model autocorrelations. We set $p_{\text{ext}} = 10^{-4}$, but the strength of external input in the linear model does not affect the autocorrelation timescales.

## Network model with different unit types
In this model, two unit-types A and B are placed at each node of a two-dimensional square lattice (Fig. 5a). The connectivity between the units is random and each unit is connected to 8 other units of any type.

The activity of each unit is given by a binary state variable $S_i \in \{0, 1\}$ with transition probabilities as in the spatial linear model (Eq. (12)), but with different probabilities for the self-excitation ($p_{\text{self},A}$, $p_{\text{self},B}$) and recurrent interactions ($p_{r,A}$, $p_{r,B}$) for each unit type. In order for both unit types to operate in the same dynamical regime, we set $p_{\text{self},A} + 8p_{r,A} = p_{\text{self},B} + 8p_{r,B} = \text{BP}$. Simulations were performed as for the spatial network, but auto- and cross-correlations were computed using the summed activity of two units $A$ and $B$ at each lattice node.

## Network model with synaptic filtering
The model operates on a two-dimensional square lattice, where each unit on the lattice is connected to 8 randomly selected units (Fig. 5b). We define the discrete-time dynamics of units in this model based on a previously proposed continuous rate model with synaptic filtering[46]. The transition probabilities for each binary unit $S_i \in \{0, 1\}$ at time-step $t'$ are governed by

$$p(S_i = 0 \rightarrow 1) = p_{\text{ext}} + f\left(\sum_j S_j\right),$$
$$p(S_i = 1 \rightarrow 0) = 1 - \left(p_{\text{ext}} + p_s + f\left(\sum_j S_j\right)\right). \tag{14}$$

Here, $f(\sum_j S_j)$ is a low-pass filter on recurrent inputs to each unit with the time constant $\tau_{\text{synapse}}$, which evolves in discrete time-steps:

$$f\left(t' + 1, \sum_j S_j\right) = f\left(t', \sum_j S_j\right) + \frac{p_r \sum_j S_j - f(t', \sum_j S_j)}{\tau_{\text{synapse}}/\Delta t'}, \tag{15}$$

where $\Delta t' = 1$ ms is the duration of each time step. Simulations and computation of auto- and cross-correlations were the same as for the spatial network.

## Analytical derivation of local timescales in the spatial network model
For analytical derivations, we derived a continuous-time rate model corresponding to the linear probabilistic network model (Eq. (12)), with the transition rates defined as

$$w(S_i = 0 \rightarrow 1) = \alpha_1 + \beta_1 \sum_j S_j,$$
$$w(S_i = 1 \rightarrow 0) = \alpha_2 - \beta_2 \sum_j S_j. \tag{16}$$

These equations contain two non-interaction terms $\alpha_1 = p_{\text{ext}}\left[\frac{-\ln(p_s)}{(1-p_s)\Delta t'}\right]$ and $\alpha_2 = (1 - p_s - p_{\text{ext}})\left[\frac{-\ln(p_s)}{(1-p_s)\Delta t'}\right]$, and two interaction terms $\beta_1 = \beta_2 = p_r\left[\frac{-\ln(p_s)}{(1-p_s)\Delta t'}\right]$, where $\Delta t' = 1$ ms is the duration of each time step (details in ref. 47). For this model, the probability of units to stay in a certain configuration $\{S\} = \{S_1, S_2, \ldots, S_N\}$ at time $t'$ is denoted as

$P(\{S\}, t')$. The master equation describing the time evolution of $P(\{S\}, t')$ is given by[50]:

$$\frac{d}{dt'} P(\{S\}, t') = -P(\{S\}, t') \sum_i w(S_i) + \sum_i P(\{S\}^{i*}, t') w(1 - S_i), \quad (17)$$

where $\{S\}^{i*} = \{S_1, S_2, \ldots, 1 - S_i, \ldots, S_N\}$. Using the master equation, we can write the time evolution for the first and second moments as

$$\frac{d}{dt'} \langle S_i \rangle(t') = \sum_{\{S\}} P(\{S\}, t')[w(S_i)(1 - 2S_i)], \quad (18)$$

$$\frac{d}{dt'} \langle S_i S_j \rangle(t') = \sum_{\{S\}} P(\{S\}, t')[w(S_i)(1 - 2S_i)S_j + w(S_j)(1 - 2S_j)S_i], \quad (19)$$

and for the time-delayed quadratic moment at time-lag $t$ as

$$\frac{d}{dt} \langle S_i(t') S_j(t' + t) \rangle = \langle S_i(t')(1 - 2S_j(t' + t)) w(S_j(t' + t)) \rangle. \quad (20)$$

By setting the right side of Eq. (18) to zero and averaging across all units, we can compute the steady-state mean activity

$$\bar{S} = \frac{1}{N} \sum_i \langle S_i \rangle = \frac{\alpha_1}{\alpha_1 + \alpha_2 - n\beta_1} = \frac{p_{\text{ext}}}{1 - (p_s + 8p_r)}, \quad (21)$$

where $n = 8$ is the number of incoming connections to each unit.

We compute the timescales analytically for the network with local connections ($r = 1$). From Eq. (20), we can derive the equation for the average autocorrelation of each unit $AC(t)$ as

$$\frac{1}{\alpha_1 + \alpha_2} \frac{d}{dt} AC(t) = -AC(t) + \frac{\beta_1}{\alpha_1 + \alpha_2} \sum_{\mathbf{x}} CC(\mathbf{x}, t). \quad (22)$$

Here $CC(\mathbf{x}, t)$ is the cross-correlation between each unit at location $(i, j)$ and its 8 nearest neighbors $\mathbf{x} = (i \pm 1, j \pm 1)$. The cross-correlation term in this equation gives rise to the interaction timescales in the autocorrelation. By neglecting the cross-correlation term, we can solve the Eq. (22) to get the self-excitation timescale

$$\tau_{\text{self}} = \frac{1}{\alpha_1 + \alpha_2} = -\frac{\Delta t'}{\ln(p_s)}. \quad (23)$$

Solving the dynamical equation for the time-delayed cross-correlation (Eq. (20)) in the Fourier domain gives the interaction timescales (Supplementary Note 4, details in[47]):

$$\tau_{\text{int},\mathbf{k}}(\mathbf{k} = (k_1, k_2)) = \frac{\tau_{\text{self}}}{1 - \frac{n}{4} \frac{\beta_1}{\alpha_1 + \alpha_2} [\cos(k_1) + \cos(k_2) + 2\cos(k_1)\cos(k_2)]}$$
$$= -\frac{\Delta t'}{\ln(p_s)} \cdot \frac{1}{1 - p_s - 2p_r[\cos(k_1) + \cos(k_2) + 2\cos(k_1)\cos(k_2)]}, \quad (24)$$

where $\mathbf{k} = (k_1, k_2)$ are the spatial frequencies in the Fourier space. For each $\mathbf{k}$ we get a different interaction timescale. Smaller $\mathbf{k}$ (low spatial frequencies) correspond to interactions on larger spatial scales, whereas larger $\mathbf{k}$ (high spatial frequencies) correspond to interactions on more local spatial scales. The largest interaction timescale (the global timescale) is defined based on the zero spatial frequency mode:

$$\tau_{\text{global}} = \tau_{\text{int},\mathbf{k}}(\mathbf{k} = (0, 0)) = \frac{1}{\alpha_1 + \alpha_2 - n\beta_1} = -\frac{\Delta t'(1 - p_s)}{(1 - p_s - 8p_r) \ln(p_s)}. \quad (25)$$

In these derivations, we defined distances between units as Euclidean distances and discarded the contributions from third and higher moments.

Considering the self-excitation and interaction (i.e. cross-correlation) terms, we can write down the analytical form of the autocorrelation function as

$$AC(t) = A \exp\left(-\frac{t}{\tau_{\text{self}}}\right) + \sum_{k_1, k_2 = 0}^{\frac{2\pi(N'/2 - 1)}{N'}} \tilde{CC}(k_1, k_2) \left[\exp\left(-\frac{t}{\tau_{\text{int},\mathbf{k}}(k_1, k_2)}\right)\right], \quad (26)$$

where $A$ is the normalization constant to get $AC(t = 0) = 1$. $N'$ is the number of units in each dimension: $N' \times N' = N$. This equation shows that the autocorrelation function contains self-excitation timescale $\tau_{\text{self}}$ and $N'^2/4$ interaction timescales weighted by the amplitude of cross-correlation function $\tilde{CC}(k_1, k_2)$ for the given spatial frequency mode $(k_1, k_2)$. We can approximate the slow decay of the autocorrelation with an effective interaction timescale $\tau_{\text{int}}$ given by the weighted average of all interaction timescales created by different spatial frequency modes[47]:

$$\tau_{\text{int}} = \sum_{k_1, k_2 = 0}^{\frac{2\pi(N'/2 - 1)}{N'}} \left[\frac{\tilde{CC}(k_1, k_2)}{CC(0, 0)}\right] \tau_{\text{int},\mathbf{k}}(k_1, k_2). \quad (27)$$

Here $CC(0, 0)$ is given by $\sum_{k_1, k_2 = 0}^{\frac{2\pi(N'/2 - 1)}{N'}} \tilde{CC}(k_1, k_2)$.

The analytical approximation of the effective interaction timescale is more accurate when the dynamics are away from the critical point. Close to the critical point ($BP \to 1$), the mean-field approximations are not valid.

The self-excitation timescale for the discrete-time network model can also be obtained analytically using the autocorrelation of a two-state Markov process driven by the self-excitation and external input. Using the transition matrix (considering the linear model)

$$\mathbb{P} = \begin{bmatrix} 1 - p_{\text{ext}} & p_{\text{ext}} \\ 1 - (p_s + p_{\text{ext}}) & p_s + p_{\text{ext}} \end{bmatrix}, \quad (28)$$

we can compute the autocorrelation of the Markov process at time-lag $t$ (Supplementary Note 3):

$$AC_{\text{2SMP}}(t) = p_s^t. \quad (29)$$

The decay timescale of this autocorrelation is equivalent to the self-excitation timescale in the network model

$$\tau_{\text{self}} = -(\ln(p_s))^{-1}, \quad (30)$$

which for $\Delta t' = 1$ is equivalent to Eq. (23).

**Analytical derivation of timescales for nonlinear interactions**

We can write down the general form of transition rates described previously in Eq. (16) as

$$\begin{aligned}
\omega(0 \to 1) &= \alpha_1 + \beta_1' \mathcal{F}\left(\sum_j S_j + I\right), \\
\omega(1 \to 0) &= \alpha_2 - \beta_2' \mathcal{F}\left(\sum_j S_j + I\right).
\end{aligned} \quad (31)$$

$\mathcal{F}(x)$ is a non-linear activation function that is a monotonically increasing function of $x$ and satisfies $\mathcal{F}(0) = 0$, $\mathcal{F}(\infty) = 1$. Here we

consider $\mathcal{F}$ of the form:

$$\mathcal{F}\left(\sum_j S_j\right) = 1 - \exp\left(-\frac{\theta}{n}\sum_j S_j\right), \qquad (32)$$

where $\theta$ is a positive constant that controls the gain of recurrent inputs, and $n$ is the number of connected neighbors to each target unit. The activation function with a constant global input current $I \geqslant 0$ can be written as:

$$\mathcal{F}(n\bar{S}+I) = 1 - \exp(-\theta\bar{S} - I), \qquad (33)$$

where $\bar{S}$ is the steady-state mean activity. Here $I$ is a constant input current that uniformly increases activation of all units, which is different from $p_{\text{ext}}$ that provides stochastic and spatially random activation of units. We interpret $I$ as the attentional input (e.g., from FEF) to V4 area.

To compute the timescales in the presence of non-linearity and external input current, we can perform Taylor expansion of the interaction terms around the mean activity $\bar{S}$

$$\beta_1'\mathcal{F}\left(\sum_j S_j + I\right) = \beta_1'\mathcal{F}'(n\bar{S}+I)\left(\sum_j S_j\right) + \beta_1'\mathcal{F}_0 = \beta_1\left(\sum_j S_j\right) + \beta_1'\mathcal{F}_0, \quad (34)$$

$$\beta_2'\mathcal{F}\left(\sum_j S_j + I\right) = \beta_2'\mathcal{F}'(n\bar{S}+I)\left(\sum_j S_j\right) + \beta_2'\mathcal{F}_0 = \beta_2\left(\sum_j S_j\right) + \beta_2'\mathcal{F}_0, \quad (35)$$

where $\mathcal{F}'$ denotes the derivative of $\mathcal{F}$ and $\mathcal{F}_0$ is defined as

$$\mathcal{F}_0 = \mathcal{F}(n\bar{S}+I) - n\bar{S}\mathcal{F}'(n\bar{S}+I) + O\left(\left[\left(\sum_j S_j\right) - n\bar{S}\right]^2\right). \qquad (36)$$

Using these expansions, we can rewrite the transition rates as

$$
\begin{aligned}
\omega(0 \to 1) &= \alpha_1^{\text{eff}} + \beta_1\sum_j S_j, \\
\omega(1 \to 0) &= \alpha_2^{\text{eff}} - \beta_2\sum_j S_j,
\end{aligned}
\qquad (37)
$$

where

$$\alpha_1^{\text{eff}} = \alpha_1 + \beta_1'\mathcal{F}_0, \quad \alpha_2^{\text{eff}} = \alpha_2 - \beta_2'\mathcal{F}_0, \qquad (38)$$

$$\beta_1 = \beta_1'\mathcal{F}'(n\bar{S}+I), \quad \beta_2 = \beta_2'\mathcal{F}'(n\bar{S}+I). \qquad (39)$$

Hence, all non-interaction and interaction terms, as well as the mean activity $\bar{S}$ depend on the external input. Consequently, the self-excitation and interaction timescales become input dependent.

The explicit form of the self-excitation timescale and the global interaction timescale are given by

$$\tau_{\text{self}} = \frac{1}{\alpha_1^{\text{eff}} + \alpha_2^{\text{eff}}} = \frac{1}{\alpha_1 + \alpha_2 + (\beta_1' - \beta_2')\mathcal{F}_0}, \qquad (40)$$

and

$$\tau_{\text{global}} = \frac{\tau_{\text{self}}}{1 - \frac{n\beta_1}{\alpha_1^{\text{eff}} + \alpha_2^{\text{eff}}}} = \frac{1}{\alpha_1 + \alpha_2 + (\beta_1' - \beta_2')\left[1 - (\theta\bar{S}+1)e^{-\theta\bar{S}-I}\right] - \beta_1'\theta e^{-\theta\bar{S}-I}}. \qquad (41)$$

When $(\beta_1' - \beta_2') < 0$, increasing the external input $I$ would lead to an increase in the mean activity and the self-excitation timescale. This conditions implies that already active units are more excitable in the

next time step compared to silent units. Moreover, if in addition to $(\beta_1' - \beta_2') < 0$, we have $-|\beta_1' - \beta_2'|\bar{S} + \beta_1' < 0$, the global timescale would also increase. Other interaction timescales increase with the input when $-|\beta_1' - \beta_2'|\bar{S} + c_1\beta_1' < 0$ ($-1 < c_1 < 1$) (details in[47]). The changes in the fast timescale are smaller than in the slow timescale and can remain undetected with the limited data amount.

## Matching the timescales of the network model to neural data

To match the timescales between the model and V4 data, we used the activity autocorrelation of one unit in the network model with local connections ($r = 1$). We searched for model parameters such that the model timescales fell within the range of timescales observed in the V4 activity, which was the mean ± s.e.m of the MAP timescale-estimates across recording sessions. We computed the range for the fast timescales from the pooled attend-in and attend-away conditions, since they were not significantly different: $\tau_{1,\text{att-away}} = \tau_{1,\text{att-in}} = 4.74 \pm 0.42$ ms. We used this range for the fast timescale in both the attend-in and attend-away conditions. For the slow timescales, we computed the ranges separately for the attend-in (averaged over covert and overt) and attend-away conditions: $\tau_{2,\text{att-away}} = 117.09 \pm 10.58$ ms, $\tau_{1,\text{att-in}} = 140.97 \pm 11.51$ ms.

We fitted the self-excitation and effective interaction timescales obtained from the autocorrelation of an individual unit's activity in the model to the fast and slow timescales of V4 data estimated from the aABC method. Using Eq. (30) and Eq. (27), we found an approximate range of parameters $p_{\text{s}}$ and $p_{\text{r}}$ that reproduce V4 timescales. Then, we performed a grid search within this parameter range to identify the model timescales falling within the range of V4 timescales during attend-away and attend-in conditions. We used model simulations for grid search since the analytical results for the effective interaction timescale are approximate. We used very long model simulations ($10^5$ time steps) to obtain unbiased autocorrelations and then estimated the model timescales by fitting a double exponential function

$$\text{AC}(t) = c_1 e^{-t/\tau_1} + (1 - c_1)e^{-t/\tau_2}, \qquad (42)$$

directly to the empirical autocorrelations. We fitted the exponential function up to the time-lag $t_m = 100$ ms, the same as used for fitting the neural data autocorrelations with the aABC method.

### Reporting summary

Further information on research design is available in the Nature Portfolio Reporting Summary linked to this article.

## Data availability

All behavioral and electrophysiological data used in this study are available on Figshare at https://doi.org/10.6084/m9.figshare.19077875.v1[72] (fixation task, FT), https://doi.org/10.6084/m9.figshare.16934326.v3[71] (attention task 1, AT1), and https://doi.org/10.6084/m9.figshare.21972911.v2[73] (attention task 2, AT2). Source data are provided with this paper.

## Code availability

The code for the timescale estimation and Bayesian model comparison with the aABC method is available as a Python package at: https://github.com/roxana-zeraati/abcTau[79]. The code for simulating network models is available at: https://github.com/roxana-zeraati/spatial-network[80].

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

## Acknowledgements

This work was supported by a Sofja Kovalevskaja Award from the Alexander von Humboldt Foundation, endowed by the Federal Ministry of Education and Research (R.Z., A.L.), SMARTSTART2 program provided by Bernstein Center for Computational Neuroscience and Volkswagen Foundation (R.Z.), the NIH grant R01 EB026949 (T.A.E.), the Swartz Foundation (Y.S.), the Pershing Square Foundation (T.A.E.), the Sloan Research Fellowship (Y.S., T.A.E.), NIH grant RF1DA055666 (Y.S., T.A.E.), the NIH grant EY014924 (T.M.), the MRC grant MR/P013031/1 (M.A.G., A.T.). This work was performed with assistance from the NIH Grant S10OD028632-0. We acknowledge the support from the BMBF through the Tübingen AI Center (FKZ: 01IS18039B) and the International Max Planck Research School for the Mechanisms of Mental Function and Dysfunction (IMPRS-MMFD). A.L. is a member of the Machine Learning Cluster of Excellence, EXC number 2064/1 - Project number 39072764. We thank Julia Wang for providing the code for estimating receptive fields.

## Author contributions

R.Z., A.L., and T.A.E. designed the study. N.A.S., M.A.G., A.T., and T.M. designed the experiments. N.A.S. and M.A.G. performed the

experiments and spike sorting. R.Z., Y.S., A.L., and T.A.E. developed the analysis methods and mathematical models. R.Z. analyzed the data and performed model simulations. Y.S. performed the analytical calculations for the network model. R.Z., Y.S., N.A.S., M.A.G., A.T., T.M., A.L., and T.A.E. discussed the findings and wrote the paper.

## Competing interests

The authors declare no competing interests.
