## [Peer Review File · Nature Communications]

Intrinsic timescales in the visual cortex change with selective attention and reflect spatial connectivityReviewers' Comments:

Reviewer #1:

Remarks to the Author:

This study by Zeraati et al explores how attention modulates the timescale of neural activity in V4. Recent work has argued that different parts of the brain have different intrinsic timescales; this study tests whether the intrinsic timescale within a given area can be quickly modulated by cognitive demands. The authors find that V4 activity fluctuates on both fast and slow timescales, defined by assessment of multiunit autocorrelation functions. Attention causes an increase in the slow timescale component. The authors explore a model which can recapitulate the experimental observations. In the model, the fast timescale arises from the dynamics of self-excitation (related to the transition probabilities of a Markov process); slow timescales arise from interactions between units. The increase in slow timescale activity in the data can be recapitulated by increasing the strength of the recurrent component.

The work is clearly presented and well done. The analysis is convincing and the changes induced by attention compelling. The modeling work seems fairly straightforward. Despite these strengths, I had difficulty understanding the impact of these findings. I also had a few technical concerns.

1. The finding that attention enhances slow timescale fluctuations seems contrary to much previous work. Most relevant, the Mitchell et al (2019, Neuron) showing that attention suppresses low frequency oscillation in cortex and work by Fries showing that attention enhances gamma (high frequency) activity. Those papers refer to a larger prior literature showing suppression of low frequency activity with visual input (e.g. work of Berger), relevant for the authors comparison between spontaneous and driven activity. The measurements here are somewhat different and perhaps that explains the apparent discrepancy. But the authors should discuss extensively the differences between their conclusions and this prior work.

2. The analysis is performed on multiunit activity pooled across the probe. There was little explanation for why this was done, but it raises a substantial concern that the different timescales might reflect different neurons. Changes in timescale with attention might then reflect how different single units contribute to the pooled MUA. It seems critical to show that analysis of well-isolated single units would yield the same outcomes. Instead, the authors report similar results in the cross-correlation functions computed across channels, which doesn't address this concern adequately (or, without further explanation, at all). It seems more straightforward, and more compelling, to do the main analyses on single units isolated from each probe.

3. The model is sensible and provides a good match to the data. But this does not seem particularly difficult or insightful. Ideally, the study would compare several models and indicate which can account for the data and which cannot. In the current case, the model recapitulates the data rather than the data adjudicating between different models. Because of this, one cannot have much confidence that the parameters explored in the model are particularly important or meaningful (other models with different architectures or parameters might perform equally well, for instance modulation of the biophysical properties of individual neurons due to neuromodulation). Of course, one can always wonder whether another model would be better but modeling is most helpful when it rules out certain classes of models in addition to mimicking data. In this case, this might be difficult to do because the data are not particularly rich (i.e., there is just a measurement of a single quantity, the population autocorrelation function which doesn't strongly constrain the possible mechanisms).

4. Pg 14, par 3, last sentence: "Therefore, comparing the timescales.." This claim is much too strong. Just because this is true for this model, does not mean this is true in the brain (i.e. that other mechanisms, not related to network connectivity, dominate).

Reviewer #2:

Remarks to the Author:

This manuscript reports a study of timescales of neural dynamics using a combination of data analysis and computational modelling.

A first part of the manuscript reports an analysis of electrophysiological data recorded in monkeys performing an attention-based task. The main result of that part is that the dynamics exhibit two clearly separated timescales, a fast and a slow one. In particular, the slow timescale varies by about 20% between the attentional states of the animal.

A second part of the manuscript reports an analysis of dynamics in a recurrent network model with spatial connectivity. The core of that part is a mathematical analysis of the impact of different network parameters on dynamical timescales. Based on these insights, the manuscript argues that the modulation of the timescales in the experimental data is based on the modulation of connectivity.

Each of the two parts of the manuscript reports interesting analyses and results that should be published. The mathematical analyses seem particularly novel, and would merit being presented in more detail. However, the link between the two parts appears to be weak, and as a consequence the main claims do not seem justified. Moreover, the chosen framing of the results is unconvincing and does not take into account previous work on the topic.

Main Comments:

1. General framing: the stated goal of the paper is to determine "whether the timescales can adjust rapidly and selectively to the demands of the task". There are several important issues here:
 - the abstract and the introduction state that it is unknown whether this is possible (abstract: "it is unknown whether the timescales can adjust rapidly and selectively to the demands of a cognitive task"; intro: "These observations suggest that timescales of neural activity are hardwired in the connectivity structure and thus are fixed characteristics of the dynamics in each brain area."). A number of previous works have however reported and analysed the modulation of dynamical timescales depending on task demands. The recent, very detailed series of studies by the Jazayeri lab in particular comes to mind, see eg Wang et al 2018 "Flexible timing by temporal scaling of cortical responses". These studies clearly demonstrated that timescales can be adjusted depending on the task requirements, and suggested potential mechanism.
 - "demands of the task": it is unclear how the spatial attention task performed by the monkeys requires a modulation of timescales. This raises the question of whether the observed modulation has at all a functional role.
 - the fact that the adjustment of timescales is "rapid" is stated but not shown.
2. Disconnect between the two parts: the network model does not model the attention task performed by the monkeys, it therefore cannot address the main question of how the modulation of timescales is related to task demands. In fact, the model does not even explicitly implement attentional modulation, but assumes that top-down inputs (not included in the model) modulate recurrent interactions through an unspecified mechanism. This is a major limitation of the overall approach, and raises questions about possible conclusions.
3. Interpretation of the analysis of the network model: the analysis clearly demonstrates that changing the recurrent connections modulates the slow timescale of the dynamics. The manuscript however argues the converse: "comparing the timescales of local and global dynamics can reveal the underlying network structure to distinguish between random and spatially organized connectivity" (p.9). This reverse inference is incorrect, the modulation of the timescales cannot be used to reveal connectivity structure. This modulation can be a consequence of many other mechanisms, for example reflect the temporal dynamics of inputs received by the network, or a modulation of the gain through constant external inputs (mechanism proposed in Wang et al 2018 mentioned above). And even when modulation is a consequence of connectivity structure, it cannot be used as evidence of spatial structure rather than other types of structure leading to eigenvalues close to instability.

Again, I believe the details of the analysis are interesting, but I encourage the authors to revise the overall framing and main claims.

Reviewer #3:

Remarks to the Author:

This paper reports findings from monkey V4 during spatial attention tasks. The idea is to characterize the intrinsic time scales of the population and note how attention modulates the collection of time constants. Time constants are estimated by using a sophisticated method to estimate time constants from the autocorrelation function of spikes aggregated over sixteen channels of a microelectrode presumably sampling different parts of a cortical column. The authors identify a fast and a slow timescale; the fast time scale (on the order of a few milliseconds) is unaffected by whether or not the animal is attending to the receptive field of the column; the slower time scale (~50 ms) is modulated by attention.

This paper addresses an important and timely topic and is technically very sound. The method for estimating time constants is vastly superior to many other approaches that have been used in the literature. However, I have some concerns about the choices made in constructing the analyses. These choices reflect a priori assumptions made by the experimenters. These choices, at minimum, need to be made explicit so readers can be aware of them and incorporate these into their understanding of the results. More broadly, I encourage the authors to seriously consider these concerns; most of them could be addressed with additional analyses.

* Number of time scales.

The authors spend a great deal of effort justifying that there is more than one time scale contributing to the ACF. This argument is compelling and well-justified. However, there is no effort put into establishing that there are not more time scales. Much recent work suggests a continuous distribution of time scales in sensory regions (Latimer, et al., *J Neuro*, 2019), during working memory (Cavanagh, et al, *Nat Comm*, 2018; review in *Frontiers* 2021), and in navigation and memory in the entorhinal cortex (Tsao, et al, *Nature*, 2018; Bright, Meister et al., *PNAS* 2020). Would the method used here be able to identify if there were, in actuality, three time scales? How about if the time scales were distributed according to some smooth function? There is no question that in the presence of noise the goodness of fit that comes from adding an additional parameter would not meet traditional thresholds for accepting the parameter. But placing the null hypothesis in this way is a theoretical choice, not one required by the data.

* Range of time constants.

More effort should be put into justifying 100 ms as the upper limit of time constants considered. There is evidence that stimuli can be decoded from V1 activity more than a second in the past (e.g., Nikolic, et al. *PLOS* 2009). behavioral RTs show very slow fluctuations across several trials (in humans it has been argued that RTs can show scale-free fluctuations under some conditions). How was 100 ms determined? Do the results change as the upper limit is systematically manipulated? Presumably as the time scale becomes much larger it eventually encompasses the response, or additional trials, or the large scale structure of the experiment.

* Treating the column as a unit rather than the neuron.

This paper aggregates spikes over multiple simultaneously recorded channels from the same electrode. The authors write:

``Previous laminar recordings showed that the neural activity is synchronized across cortical layers alternating spontaneously between synchronous phases of high and low firing rates. Therefore, we pooled the spiking activity across all layers (Fig. 1d) to obtain more accurate estimates of the spike-count autocorrelations.'`

Is the argument that it's impractical to assess time constants using this method without aggregating

over units? Is there some reason a priori to believe that all neurons in a column should have the same characteristic time constants (see above)? At minimum, I'd want to see supplementary figures with ACFs and an estimate of time constants (ideally from this method, but there are other methods if that's impractical) from individual neurons. Ideally, it would be very nice to establish that the individual neurons in a column are or are not representative of the column per se (this may be very challenging).

* Connection to previous work on oscillations and synchrony

I was expecting to see more discussion of how the fast and slow time scales may connect to what we already know about oscillations in the LFP and synchrony. Perhaps the change in τ_2 is related to a change in the frequency of, or synchrony of spikes to, an ongoing oscillation with period on the order of $1/\tau_2$? Knowing this would contextualize the results in this paper with previous work in monkey neurophysiology and potentially increase the impact of the paper.

Point-by-point responses to reviewers' comments

Reviewer #1:

This study by Zeraati et al explores how attention modulates the timescale of neural activity in V4. Recent work has argued that different parts of the brain have different intrinsic timescales; this study tests whether the intrinsic timescale within a given area can be quickly modulated by cognitive demands. The authors find that V4 activity fluctuates on both fast and slow timescales, defined by assessment of multiunit autocorrelation functions. Attention causes an increase in the slow timescale component. The authors explore a model which can recapitulate the experimental observations. In the model, the fast timescale arises from the dynamics of self-excitation (related to the transition probabilities of a Markov process); slow timescales arise from interactions between units. The increase in slow timescale activity in the data can be recapitulated by increasing the strength of the recurrent component.

The work is clearly presented and well done. The analysis is convincing and the changes induced by attention compelling. The modeling work seems fairly straightforward. Despite these strengths, I had difficulty understanding the impact of these findings. I also had a few technical concerns.

We thank the reviewer for the overall positive evaluation and constructive feedback that helped strengthen our results. In particular, the suggestion to compare alternative network models led to new analyses which solidified our conclusion about the relationship between timescales and spatial network structure. Moreover, the reviewer's comments motivated us to present the details of our analytical results in a separate theory paper (Shi et al., arXiv 2022, under review), which establishes a nontrivial relationship between timescales and spatial connectivity, providing the foundation for our data analysis here. Finally, our new finding that the slow timescales predict behavioral performance increases the impact of this work (new Fig. 4). We provide detailed responses below. The line numbers refer to the lines in the revised manuscript without track changes.

1. The finding that attention enhances slow timescale fluctuations seems contrary to much previous work. Most relevant, the Mitchell et al (2019, Neuron) showing that attention suppresses low frequency oscillation in cortex and work by Fries showing that attention enhances gamma (high frequency) activity. Those papers refer to a larger prior literature showing suppression of low frequency activity with visual input (e.g. work of Berger), relevant for the authors comparison between spontaneous and driven activity. The measurements here are somewhat different and perhaps that explains the apparent discrepancy. But the authors should discuss extensively the differences between their conclusions and this prior work.

Following the reviewer's suggestion, we performed additional analyses to investigate the relationship between the increase of slow timescales and the reduction in the power of low-frequency fluctuations during attention. We found that the two phenomena are consistent with each other. First, we observe a reduction in low-frequency and an increase in gamma-range power of local field potentials (LFPs) during attention in our data (attention task 1, Steinmetz and Moore, 2014), consistent with previous reports (Fries et al., 2001; Chalk et al., 2010; Ferro et al., 2021). We also observe a reduction at low frequencies in the power spectral density (PSD) of spiking activity in our data (new Supplementary Fig. 6) as in previous studies (Mitchell et al., 2009). To test how these changes in the PSD shape are related to the modulation of timescales during attention, we estimated timescales directly from PSDs using our aABC method in the frequency domain (Zeraati et al., 2022). We fitted the PSD of spiking activity with the two-timescale generative model and obtained the same results as with the time-domain analyses: the fast timescale did not change and the slow timescale increased with attention (new Supplementary Fig. 7, cf. Fig. 3b,e).

These results show that the increase in the slow timescale is consistent with the reduction in the power of low-frequency fluctuations during attention.

To explain the co-occurrence of these two phenomena, we considered theoretically how the PSD shape changes under the modulation of timescales. In the frequency domain, PSD of a one-timescale process with an exponential autocorrelation is a Lorentzian function $\frac{c}{f^2 + f_k^2}$, where f is the frequency, f_k is the knee frequency related to the timescale τ as $f_k = (2\pi\tau)^{-1}$, and c is a normalization constant. In logarithmic-logarithmic coordinates this function has a plateau at low frequencies transitioning to the $1/f^2$ decay at the knee frequency. The PSD of a two-timescale process is a mixture of two Lorentzian functions, one with a large knee frequency (fast timescale) and one with a small knee frequency (slow timescale). In V4 data, the slow timescale increased and the fast timescale did not change during attention. When the slow timescale increases, the small knee frequency shifts to even lower frequencies. At the same time, the large knee frequency (the fast timescale) does not change, keeping the power at high frequencies constant. For the values of timescales estimated in V4, these effects together produce a reduction in the low-frequency power (~2-20Hz) (new Supplementary Fig. 6). The decrease in the small knee frequency indeed results in an increase of power at very low frequencies (<2Hz), which, however, is not detectable with short observation windows (<500ms) used in previous studies. With a longer observation window of 700ms, we indeed see a PSD increase at a very low frequency (~1.4Hz) during attention (Supplementary Fig. 6). We describe these results in the Results (lines 164-166), new Supplementary Note 1, and in new Supplementary Figs. 6,7.

2. The analysis is performed on multiunit activity pooled across the probe. There was little explanation for why this was done, but it raises a substantial concern that the different timescales might reflect different neurons. Changes in timescale with attention might then reflect how different single units contribute to the pooled MUA. It seems critical to show that analysis of well-isolated single units would yield the same outcomes. Instead, the authors report similar results in the cross-correlation functions computed across channels, which doesn't address this concern adequately (or, without further explanation, at all). It seems more straightforward, and more compelling, to do the main analyses on single units isolated from each probe.

The reviewer is right that, in principle, multiple timescales in autocorrelations of MUA could reflect heterogeneous timescales of different neurons, and analyses of well-isolated single-unit activity (SAU) would be ideal for addressing this concern. However, due to low firing rates, SUA often does not yield sufficient data for conclusive model comparison, therefore, we used MUA in the original manuscript. Following the reviewer's suggestion, we fitted autocorrelations of SUA during the fixation task (which had the longest trial duration of 3s and thus the largest data amount) and performed the model comparison to determine the number of timescales (new Supplementary Fig. 8). While some single units clearly showed two distinct timescales, the model comparison was inconclusive for most units because autocorrelations were dominated by noise due to low data amount. In addition, some SUA autocorrelations exhibited more complex dynamics (e.g., oscillations or refractory period) than what can be captured by the one- or two-timescale model. Due to these limitations of SUA (also reported by previous studies (Cavanagh, et al., 2016, 2018; Wasmuht et al., 2018)), it is not possible to conclusively perform our analyses of timescales and their attentional modulation using SUA.

Nevertheless, we can conclusively answer the question of whether multiple timescales in MUA reflect heterogeneous timescales of different neurons or shared population dynamics by analyzing cross-correlations. The cross-correlations of MUA across channels show the fast and slow timescales, which indicates that the dominant timescales arise from population dynamics shared by neurons within a column. To further test this idea, we analyzed how timescales of auto- and cross-correlations depend on the

lateral distance between channels (new Fig. 5). If multiple timescales in MUA merely reflected different neurons with heterogeneous biophysical properties, these timescales should not depend on distance. However, we found that the timescales of auto- and cross-correlations changed systematically with distance as predicted by the network model with spatial connectivity (see reply to point 3 below). This new result further supports the idea that multiple timescales in MUA reflect shared population dynamics and not merely heterogeneous properties of single neurons.

We describe these analyses in Results (lines 194-201), new Supplementary Note 2 and new Supplementary Fig. 8.

3. The model is sensible and provides a good match to the data. But this does not seem particularly difficult or insightful. Ideally, the study would compare several models and indicate which can account for the data and which cannot. In the current case, the model recapitulates the data rather than the data adjudicating between different models. Because of this, one cannot have much confidence that the parameters explored in the model are particularly important or meaningful (other models with different architectures or parameters might perform equally well, for instance modulation of the biophysical properties of individual neurons due to neuromodulation). Of course, one can always wonder whether another model would be better but modeling is most helpful when it rules out certain classes of models in addition to mimicking data. In this case, this might be difficult to do because the data are not particularly rich (i.e., there is just a measurement of a single quantity, the population autocorrelation function which doesn't strongly constrain the possible mechanisms).

Our main hypothesis is that multiple timescales in neural activity arise from the spatially arranged connectivity, which we implemented in the spatial network model in the original manuscript. We agree that it is insightful to also consider the alternative hypothesis that timescales arise from another mechanism different from spatial connectivity. To test for this possibility, we developed two alternative network models with random connectivity, in which multiple autocorrelation timescales arise from biophysical properties of individual neurons instead of spatial network interactions (new Fig. 5). We considered two alternative mechanisms for generating two timescales in autocorrelations: 1) mixing activity of two cell types each with a distinct (slow or fast) biophysical timescale, 2) combining two different local biophysical processes, i.e. a fast membrane time constant and a slow synaptic time constant (Beiran and Ostojic, 2019). We implemented all three mechanisms using the same modeling framework for a fair comparison. All three proposed mechanisms can account equally well for multiple timescales in V4 autocorrelations, but they make distinct predictions about cross-correlations. Specifically, the model with spatial connectivity predicts that cross-correlations become weaker and dominated by slower timescales at longer distances (analytical derivation in Methods, details in Shi et al., 2022). In contrast, two other models predict that the cross-correlations are distance-independent.

We tested these model predictions in our experimental data to determine which mechanism, spatial network interactions or local biophysical properties, is more consistent with neural dynamics in V4. We measured how timescales in cross-correlations depend on the lateral cortical distance, using distances between receptive field centers as a proxy for cortical distances. We found that timescales of V4 cross-correlations increased with lateral distance in agreement with the spatial network mechanism (new Fig. 5k, new Supplementary Fig. 9). These new results support our hypothesis that multiple timescales in V4 activity arise from spatially structured recurrent interactions in the visual cortex and not from biophysical properties of individual neurons alone. Although multiple timescales can also arise from a combination of spatial interactions and biophysical properties (e.g., synaptic timescale, new Supplementary Fig. 10), spatial connectivity is required to explain the distance-dependence of cross-correlation timescales. We describe these new findings in Results (lines 187-359), new Fig. 5, and new Supplementary Figs. 9,10.

4. Pg 14, par 3, last sentence: "Therefore, comparing the timescales.." This claim is much too strong. Just because this is true for this model, does not mean this is true in the brain (i.e. that other mechanisms, not related to network connectivity, dominate).

We agree that the reverse claim is too strong. We removed this sentence from the Discussion and replaced it with "*These results show that local temporal and spatial correlations of neural dynamics are closely tied together.*" (lines 432-433).

Reviewer #2:

This manuscript reports a study of timescales of neural dynamics using a combination of data analysis and computational modelling.

A first part of the manuscript reports an analysis of electrophysiological data recorded in monkeys performing an attention-based task. The main result of that part is that the dynamics exhibit two clearly separated timescales, a fast and a slow one. In particular, the slow timescale varies by about 20% between the attentional states of the animal.

A second part of the manuscript reports an analysis of dynamics in a recurrent network model with spatial connectivity. The core of that part is a mathematical analysis of the impact of different network parameters on dynamical timescales. Based on these insights, the manuscript argues that the modulation of the timescales in the experimental data is based on the modulation of connectivity.

Each of the two parts of the manuscript reports interesting analyses and results that should be published. The mathematical analyses seem particularly novel, and would merit being presented in more detail. However, the link between the two parts appears to be weak, and as a consequence the main claims do not seem justified. Moreover, the chosen framing of the results is unconvincing and does not take into account previous work on the topic.

We thank the reviewer for the overall positive assessment and suggestions that helped us improve the paper. We addressed all reviewer's comments in the revised manuscript, in particular, we included new analyses to strengthen the link between the model and data. We also revised the framing of our results in the context of previous work on the topic. The line numbers refer to the lines in the revised manuscript without track changes.

Main Comments:

1. General framing: the stated goal of the paper is to determine "whether the timescales can adjust rapidly and selectively to the demands of the task". There are several important issues here:

- the abstract and the introduction state that is unknown whether this is possible (abstract: "it is unknown whether the timescales can adjust rapidly and selectively to the demands of a cognitive task"; intro: "These observations suggest that timescales of neural activity are hardwired in the connectivity structure and thus are fixed characteristics of the dynamics in each brain area."). A number of previous works have however reported and analysed the modulation of dynamical timescales depending on task demands. The recent, very detailed series of studies by the Jazayeri lab in particular comes to mind, see eg Wang et al 2018 "Flexible timing by temporal scaling of cortical responses". These studies clearly demonstrated that timescales can be adjusted depending on the task requirements, and suggested potential mechanism.

We revised the general framing of our work in the abstract and introduction to emphasize our two main contributions. First, we establish analytically that spatially arranged connectivity generates multiple timescales in local population activity. We find support for this theoretical prediction in V4 recordings with

new modeling and data analysis (new Fig. 5). Second, we show that intrinsic timescales of endogenous activity fluctuations are selectively modulated during spatial attention. This result is new and differs from previous work on the modulation of “dynamical timescales” cited by the reviewer. Previous studies, including the work from Jazayeri lab, showed that the timescale of trial-average neural response can change according to the task condition. However, the timescale of trial-average response is not the same as the intrinsic timescale of activity fluctuations on single trials. To isolate intrinsic timescales, we subtract the trial-average response from the activity on each trial, thus removing the timescales associated with the trial-average response (Murray et al., 2014; Wasmuht et al., 2018; Siegle et al., 2021). These two types of timescales are dissociable and can depend differently on experimental conditions. For example, the timescales of trial-average response increase through the mouse visual cortical hierarchy, whereas the intrinsic timescales do not change (Siegle et al., 2021). In addition, for a fixed trial-average response in a specific task condition, the intrinsic timescale of neural dynamics varies substantially across trials and these changes are predictive of the reaction time in a decision-making task (Boucher et al., 2022). Therefore, previous findings that timescales of trial-average response can change according to the task condition do not imply that the intrinsic fluctuation timescales can be modulated as well. Whether intrinsic fluctuation timescales can change with temporal and spatial specificity during a cognitive task has not been previously tested. We clarified the distinction between the intrinsic and trial-average timescales in the revised Introduction (lines 43-59).

- "demands of the task": it is unclear how the spatial attention task performed by the monkeys requires a modulation of timescales. This raises the question of whether the observed modulation has at all a functional role.

To address the question about the functional significance of the observed changes in timescales, we tested whether the slow timescales predicted behavioral performance. We found that on average monkeys responded to a stimulus change faster in sessions with longer slow timescales of neurons with the receptive fields in the attended location. In other words, the average reaction time was negatively correlated with the slow timescale in the attend-in condition, but was not correlated with the slow timescale in the attend-away condition. The spatial selectivity of this effect suggests that the increase in the slow timescale may contribute to behavioral benefits of selective spatial attention. We present these new results in the Results (lines 172-186), new Fig. 4, and new Supplementary Tables 1,2.

- the fact that the adjustment of timescales is "rapid" is stated but not shown.

By “rapid” we mean modulation of timescales due to fast changes in the attention state of monkeys, from one trial to the next. For clarity, we replaced “rapid” with “trial-to-trial” changes in the revised manuscript.

2. Disconnect between the two parts: the network model does not model the attention task performed by the monkeys, it therefore cannot address the main question of how the modulation of timescales is related to task demands. In fact, the model does not even explicitly implement attentional modulation, but assumes that top-down inputs (not included in the model) modulate recurrent interactions through an unspecified mechanism. This is a major limitation of the overall approach, and raises questions about possible conclusions.

Our approach of modeling attentional modulation in the visual cortex without explicitly modeling the full task execution by other cortical areas is standard in the literature (Huang et al., 2019, 2022; Kanashiro et al., 2017; Rabinowitz et al., 2015; Reynolds & Heeger, 2009; Ardid et al., 2007, 2010). In line with these studies, we focus on attention-related changes in sensory representations rather than on dynamics of downstream areas driving stimulus selection and motor response generation. Complementary to our approach, a recent modeling work tested how attentional modulation of neural activity affects task performance in a multi-layer network (Lindsey & Miller, 2018) by manually applying attention-like changes to neural activity without considering mechanisms driving these changes. We agree that developing an

integrated multi-area model of attention that performs the task and accounts for mechanisms of attention-related changes in the sensory cortex is an important goal for future research, beyond the scope of this work.

To address the behavioral relevance of attentional modulation of timescales, we analyzed the relationship between timescales and monkeys' reaction times (see reply to point 1). We found that slow timescales were predictive of reaction times, selectively in the attended location. These new findings support the behavioral significance of observed changes in intrinsic timescales during attention.

The question about a mechanism underlying changes in the strengths of recurrent interactions is definitely important, and we addressed it further by incorporating non-linear recurrent interactions in the network model. In the original manuscript, the recurrent interactions in the model were linear, hence, their strengths did not depend on the external input. The original model identified changes in the linear interaction strengths necessary to account for the observed modulation of timescales, which can be mediated by biophysical processes such as neuromodulation, consistent with experimental observations (Thiele & Bellgrove, 2018). In the revised manuscript, we provided new analytical derivations in the model with non-linear recurrent interactions which show that the effective strengths of recurrent interactions can also change by external input (Methods, details in Shi et al. 2022). The input alters the operating regime of network dynamics changing the effective strength of recurrent interactions. We show that with non-linear interactions, timescales can be modulated by the input to the network, e.g., top-down inputs during attention. Similar input-dependent mechanisms were suggested previously in networks with non-linear interactions (Hennequin et al. 2018, Wang et al., 2018). We describe these new analyses in Results (lines 386-391), Discussion (lines 477-494), and Methods (lines 836-862).

3. Interpretation of the analysis of the network model: the analysis clearly demonstrates that changing the recurrent connections modulates the slow timescale of the dynamics. The manuscript however argues the converse: "comparing the timescales of local and global dynamics can reveal the underlying network structure to distinguish between random and spatially organized connectivity" (p.9). This reverse inference is incorrect, the modulation of the timescales cannot be used to reveal connectivity structure. This modulation can be a consequence of many other mechanisms, for example reflect the temporal dynamics of inputs received by the network, or a modulation of the gain through constant external inputs (mechanism proposed in Wang et al 2018 mentioned above). And even when modulation is a consequence of connectivity structure, it cannot be used as evidence of spatial structure rather than other types of structure leading to eigenvalues close to instability.

We agree that the reverse claim is too strong. We removed this sentence from Discussion and replaced it with "*These results show that local temporal and spatial correlations of neural dynamics are closely tied together.*" (lines 432-433).

Reviewer #3:

This paper reports findings from monkey V4 during spatial attention tasks. The idea is to characterize the intrinsic time scales of the population and note how attention modulates the collection of time constants. Time constants are estimated by using a sophisticated method to estimate time constants from the autocorrelation function of spikes aggregated over sixteen channels of a microelectrode presumably sampling different parts of a cortical column. The authors identify a fast and a slow timescale; the fast time scale (on the order of a few milliseconds) is unaffected by whether or not the animal is attending to the receptive field of the column; the slower time scale (~50 ms) is modulated by attention.

This paper addresses an important and timely topic and is technically very sound. The method for estimating time constants is vastly superior to many other approaches that have been used in the literature. However, I have some concerns about the choices made in constructing the analyses. These choices reflect a priori assumptions made by the experimenters. These choices, at minimum, need to be made explicit so readers can be aware of them and incorporate these into their understanding of the results. More broadly, I encourage the authors to seriously consider these concerns; most of them could be addressed with additional analyses.

We thank the reviewer for this generous assessment and constructive feedback that helped us strengthen the paper. We addressed all comments in the revised manuscript with new analyses. The line numbers refer to the lines in the revised manuscript without track changes.

* Number of time scales.

The authors spend a great deal of effort justifying that there is more than one time scale contributing to the ACF. This argument is compelling and well-justified. However, there is no effort put into establishing that there are not more time scales. Much recent work suggests a continuous distribution of time scales in sensory regions (Latimer, et al., J Neuro, 2019), during working memory (Cavanagh, et al, Nat Comm, 2018; review in Frontiers 2021), and in navigation and memory in the entorhinal cortex (Tsao, et al, Nature, 2018; Bright, Meister et al., PNAS 2020). Would the method used here be able to identify if there were, in actuality, three time scales? How about if the time scales were distributed according to some smooth function? There is no question that in the presence of noise the goodness of fit that comes from adding an additional parameter would not meet traditional thresholds for accepting the parameter. But placing the null hypothesis in this way is a theoretical choice, not one required by the data.

The previous work cited by the reviewer showed the heterogeneity of timescales across single neurons, with a continuous distribution of timescales across the population (Cavanagh, et al., 2016, 2018; Wasmuht et al., 2018; Tsao, et al, Nature, 2018; Bright, Meister et al., 2020). In these studies, each single neuron was characterized by a single timescale. In contrast, we find multiple timescales in the local population activity, and our new analyses indicate that these multiple timescales reflect shared population dynamics and not merely heterogeneous timescales of single cells (see reply to point 2 by reviewer 1). Specifically, we find multiple timescales in autocorrelations of example single units with sufficient data (new Supplementary Fig. 8). Moreover, timescales of cross-correlations in our data increase with lateral cortical distance confirming predictions of the spatial network model, and this effect cannot be explained by models with random connectivity (new Fig. 5). Thus, our main contribution is to show the existence of more than one timescale in the local population activity and relate the timescales to spatial network interactions. Note that Latimer et al. 2019 described responses of single neurons to sensory stimuli using filters with multiple timescales, however, such stimulus-induced timescales are distinct from intrinsic timescales that we study (see reply to point 1 by reviewer 2).

Our network model predicts that spatially arranged connectivity generates multiple timescales in auto- and cross-correlations. We show analytically that auto- and cross-correlations contain multiple interaction timescales arising from fluctuations at different spatial frequency modes (Methods, details in Shi et al. 2022). The ability to detect multiple timescales in autocorrelations depends on the length of observations, since different timescales dominate the autocorrelation over different ranges of time lags (Fig. 5 in Shi et al., 2022). Thus, any conclusion about the exact number of timescales in empirical autocorrelations is necessarily confounded by the observation length and available data amount. In addition, for a finite observation length, multiple interaction timescales in the model are accurately approximated by a single effective interaction timescale (Figs. 5f, 7c). Therefore, we made a pragmatic decision to convincingly show that neural activity contains more than one timescale rather than determining the number of timescales discernable from the available data.

Our aABC method can in principle estimate any number of timescales when enough data are available (Zeraati et al. 2022). Following the reviewer's suggestion, we used the aABC method to compare the two- and three-timescale models fitted to the data from the fixation task which had the longest trial duration of 3s. We found that with this data amount, the three-timescale model did not provide a better fit than the two-timescale model (new Supplementary Fig. 3). Thus, the two-timescale model provided a parsimonious description of neural dynamics in our data for the given observation length. Future experiments with longer trial durations may enable detecting additional timescales in neural activity. We describe these findings in Results (lines 138-141) and new Supplementary Fig. 3. We also added the point that future models can incorporate the heterogeneity of timescales in the Discussion (lines 498-500).

* Range of time constants.

More effort should be put into justifying 100 ms as the upper limit of time constants considered. There is evidence that stimuli can be decoded from V1 activity more than a second in the past (e.g., Nikolic, et al. PLOS 2009). behavioral RTs show very slow fluctuations across several trials (in humans it has been argued that RTs can show scale-free fluctuations under some conditions). How was 100 ms determined? Do the results change as the upper limit is systematically manipulated? Presumably as the time scale becomes much larger it eventually encompasses the response, or additional trials, or the large scale structure of the experiment.

We did not impose the upper limit of 100ms on the estimated timescales. The timescales estimated from V4 data range well above 100ms, with some timescales larger than 400ms (Fig. 2c). The number 100ms is the maximum time-lag t_m used for fitting autocorrelations in the time domain. It defines the range over which we compute the distance between the autocorrelation of the neural data and synthetic data from the generative model. The fitting range does not limit the range of timescales estimated with the aABC method. The fitting range excludes noisy values in the autocorrelation tail and thus reduces noise in measured distances, which helps the aABC algorithm to converge faster. Using synthetic data with known ground-truth timescales, we showed that the fitting range does not affect the accuracy of estimated timescales and only changes the width of the estimated posteriors (Supplementary Fig. 11 in Zeraati et al., 2022).

Prompted by the reviewer's comment, we performed additional control analyses to show that the fitting range did not affect our estimation of V4 timescales. We estimated timescales by fitting the whole shape of power spectral density in the frequency domain. The frequency-domain fitting converges faster in wall-clock time than time-domain fitting (Zeraati et al. 2022). The results of these new analyses were in agreement with the time-domain fits with a limited fitting range (new Supplementary Fig. 7, cf. Fig. 3). We describe this control analysis in Methods (lines 646-651), and new Supplementary Fig. 7.

We note that we computed autocorrelations from ongoing dynamics within each trial. We subtracted trial-specific means to remove slow changes in firing rate across trials (Methods, lines 568-583), such as slow drift related to changes in the arousal or motivation state (Cowley et al. 2020). Thus, the range of timescales was limited to the trial duration, focusing on timescales related to endogenous network dynamics and their attentional modulation.

* Treating the column as a unit rather than the neuron.

This paper aggregates spikes over multiple simultaneously recorded channels from the same electrode. The authors write:

"Previous laminar recordings showed that the neural activity is synchronized across cortical layers alternating spontaneously between synchronous phases of high and low firing rates. Therefore, we pooled the spiking activity across all layers (Fig. 1d) to obtain more accurate estimates of the spike-count autocorrelations."

Is the argument that it's impractical to assess time constants using this method without aggregating over units? Is there some reason a priori to believe that all neurons in a column should have the same

characteristic time constants (see above)? At minimum, I'd want to see supplementary figures with ACFs and an estimate of time constants (ideally from this method, but there are other methods if that's impractical) from individual neurons. Ideally, it would be very nice to establish that the individual neurons in a column are or are not representative of the column per se (this may be very challenging).

We agree that it would be ideal to analyze timescales using well-isolated single-unit activity (SAU). However, due to low firing rates, SUA often does not yield sufficient data for conclusive model comparison, therefore, we used MUA in the original manuscript. Prompted by the reviewer's comment, we fitted autocorrelations of SUA during the fixation task (which had the longest trial duration of 3s and thus the largest data amount) and performed the model comparison to determine the number of timescales (new Supplementary Fig. 8). While some single units clearly showed two distinct timescales, the model comparison was inconclusive for most units because autocorrelations were dominated by noise due to low data amount. In addition, some SUA autocorrelations exhibited more complex dynamics (e.g., oscillations or refractory period) than what can be captured by the one- or two-timescale model. Due to these limitations of SUA (also reported by previous studies (Cavanagh, et al., 2016, 2018; Wasmuht et al., 2018)), it is not possible to conclusively perform our analyses of timescales and their attentional modulation using SUA.

Thus, while some single neurons show clear evidence of multiple timescales as in the population activity, SUA definitely has greater complexity and heterogeneity than MUA. Nevertheless, we show that multiple timescales in MUA reflect shared population dynamics and not merely heterogeneous timescales of different neurons (see reply to point 2 of reviewer 1). In particular, timescales of cross-correlations in our data increase with lateral cortical distance confirming predictions of the spatial network model, and this effect cannot be explained by single-cell heterogeneity alone (new Fig. 5).

We describe these analyses in Results (lines 194-201), new Supplementary Note 2 and new Supplementary Figure 8.

* Connection to previous work on oscillations and synchrony

I was expecting to see more discussion of how the fast and slow time scales may connect to what we already know about oscillations in the LFP and synchrony. Perhaps the change in τ_2 is related to a change in the frequency of, or synchrony of spikes to, an ongoing oscillation with period on the order of $1/\tau_2$? Knowing this would contextualize the results in this paper with previous work in monkey neurophysiology and potentially increase the impact of the paper.

Following the reviewer's suggestion (also raised in point 1 by reviewer #1), we performed additional analyses to investigate the relationship between changes in slow timescales and in the spectral power in the frequency domain. We found that our observations about timescales are compatible with previous findings about changes in the spectral power of V4 activity (e.g., LFPs) during attention. First, we observe a reduction in low-frequency and an increase in gamma-range power of local field potentials (LFPs) during attention in our data (attention task 1, Steinmetz and Moore, 2014), consistent with previous reports (Fries et al., 2001; Chalk et al., 2010; Ferro et al., 2021). We also observe a reduction at low frequencies in the power spectral density (PSD) of spiking activity in our data (new Supplementary Fig. 6) as in previous studies (Mitchell et al., 2009). To test how these changes in the PSD shape are related to modulation of timescales during attention, we estimated timescales directly from PSD using our aABC method in the frequency domain (Zeraati et al., 2022). We fitted the PSD of spiking activity with the two-timescale generative model and obtained the same results as with the time-domain analyses: the fast timescale did not change and the slow timescale increased with attention (new Supplementary Fig. 7, cf. Fig. 3b,e). These results show that the increase in the slow timescale is consistent with the reduction in the power of low-frequency fluctuations during attention.

To explain the co-occurrence of these two phenomena, we considered theoretically how the PSD shape changes under the modulation of timescales. In the frequency domain, PSD of a one-timescale process with an exponential autocorrelation is a Lorentzian function $\frac{c}{f^2 + f_k^2}$, where f is the frequency, f_k is the knee frequency related to the timescale τ as $f_k = (2\pi\tau)^{-1}$, and c is a normalization constant. In logarithmic-logarithmic coordinates this function has a plateau at low frequencies transitioning to the $1/f^2$ decay at the knee frequency. PSD of a two-timescale process is a mixture of two Lorentzian functions, one with a large knee frequency (fast timescale) and one with a small knee frequency (slow timescale). In V4 data, the slow timescale increased and the fast timescale did not change during attention. When the slow timescale increases, the small knee frequency shifts to even lower frequencies. At the same time, the large knee frequency (the fast timescale) does not change, keeping the power at high frequencies constant. For the values of timescales estimated in V4, these effects together produce a reduction in the low-frequency power (~ 2 -20Hz) (new Supplementary Fig. 6). The decrease in the small knee frequency indeed results in an increase of power at very low frequencies (< 2 Hz), which, however, is not detectable with short observation windows (< 500 ms) used in previous studies. With a longer observation window of 700ms, we indeed see a PSD increase at a very low frequency (~ 1.4 Hz) during attention (new Supplementary Fig. 6). We describe these results in the Results (lines 164-166), new Supplementary Note 1, and in new Supplementary Figs. 6,7.

References

- Ardid, S., & Wang, X.-J. (2013). A Tweaking Principle for Executive Control: Neuronal Circuit Mechanism for Rule-Based Task Switching and Conflict Resolution. *Journal of Neuroscience*, 33(50), 19504–19517. <https://doi.org/10.1523/JNEUROSCI.1356-13.2013>.
- Ardid, S., Wang, X.-J., & Compte, A. (2007). An Integrated Microcircuit Model of Attentional Processing in the Neocortex. *Journal of Neuroscience*, 27(32), 8486–8495. <https://doi.org/10.1523/JNEUROSCI.1145-07.2007>.
- Beiran, M., Meirhaeghe, N., Sohn, H., Jazayeri, M., & Ostojic, S. (2021). Parametric control of flexible timing through low-dimensional neural manifolds (p. 2021.11.08.467806). *bioRxiv*. <https://doi.org/10.1101/2021.11.08.467806>.
- Beiran, M., & Ostojic, S. (2019). Contrasting the effects of adaptation and synaptic filtering on the timescales of dynamics in recurrent networks. *PLOS Computational Biology*, 15(3), e1006893. <https://doi.org/10.1371/journal.pcbi.1006893>.
- Boucher, P. O., Wang, T., Carceroni, L., Kane, G., Shenoy, K. V., & Chandrasekaran, C. (2022). Neural population dynamics in dorsal premotor cortex underlying a reach decision (p. 2022.06.30.497070). *bioRxiv*. <https://doi.org/10.1101/2022.06.30.497070>.
- Bright, I. M., Meister, M. L. R., Cruzado, N. A., Tiganj, Z., Buffalo, E. A., & Howard, M. W. (2020). A temporal record of the past with a spectrum of time constants in the monkey entorhinal cortex. *Proceedings of the National Academy of Sciences*, 117(33), 20274–20283. <https://doi.org/10.1073/pnas.1917197117>.
- Cavanagh, S. E., Hunt, L. T., & Kennerley, S. W. (2020). A Diversity of Intrinsic Timescales Underlie Neural Computations. *Frontiers in Neural Circuits*, 14. <https://doi.org/10.3389/fncir.2020.615626>.
- Cavanagh, Sean E., John P. Towers, Joni D. Wallis, Laurence T. Hunt, and Steven W. Kennerley. (2018). Reconciling Persistent and Dynamic Hypotheses of Working Memory Coding in Prefrontal Cortex. *Nature Communications* 9 (1): 3498. <https://doi.org/10.1038/s41467-018-05873-3>.

- Cavanagh, S. E., Wallis, J. D., Kennerley, S. W., & Hunt, L. T. (2016). Autocorrelation structure at rest predicts value correlates of single neurons during reward-guided choice. *ELife*, 5, e18937. <https://doi.org/10.7554/eLife.18937>.
- Chalk, M., Herrero, J. L., Gieselmann, M. A., Delicato, L. S., Gotthardt, S., & Thiele, A. (2010). Attention Reduces Stimulus-Driven Gamma Frequency Oscillations and Spike Field Coherence in V1. *Neuron*, 66(1), 114–125. <https://doi.org/10.1016/j.neuron.2010.03.013>.
- Cowley, B. R., Snyder, A. C., Acar, K., Williamson, R. C., Yu, B. M., & Smith, M. A. (2020). Slow Drift of Neural Activity as a Signature of Impulsivity in Macaque Visual and Prefrontal Cortex. *Neuron*, 108(3), 551-567.e8. <https://doi.org/10.1016/j.neuron.2020.07.021>.
- Engel, T. A., & Steinmetz, N. A. (2019). New perspectives on dimensionality and variability from large-scale cortical dynamics. *Current Opinion in Neurobiology*, 58, 181–190. <https://doi.org/10.1016/j.conb.2019.09.003>.
- Engel, T. A., Steinmetz, N. A., Gieselmann, M. A., Thiele, A., Moore, T., & Boahen, K. (2016). Selective modulation of cortical state during spatial attention. *Science*, 354(6316), 1140–1144. <https://doi.org/10.1126/science.aag1420>.
- Ferro, D., van Kempen, J., Boyd, M., Panzeri, S., & Thiele, A. (2021). Directed information exchange between cortical layers in macaque V1 and V4 and its modulation by selective attention. *Proceedings of the National Academy of Sciences*, 118(12), e2022097118. <https://doi.org/10.1073/pnas.2022097118>
- Fries, P., Reynolds, J. H., Rorie, A. E., & Desimone, R. (2001). Modulation of Oscillatory Neuronal Synchronization by Selective Visual Attention. *Science*, 291(5508), 1560–1563. <https://doi.org/10.1126/science.1055465>.
- Hennequin, G., Ahmadian, Y., Rubin, D. B., Lengyel, M., & Miller, K. D. (2018). The Dynamical Regime of Sensory Cortex: Stable Dynamics around a Single Stimulus-Tuned Attractor Account for Patterns of Noise Variability. *Neuron*, 98(4), 846-860.e5. <https://doi.org/10.1016/j.neuron.2018.04.017>.
- Huang, C., Pouget, A., & Doiron, B. (2022). Internally generated population activity in cortical networks hinders information transmission. *Science Advances*, 8(22), eabg5244. <https://doi.org/10.1126/sciadv.abg5244>.
- Huang, C., Ruff, D. A., Pyle, R., Rosenbaum, R., Cohen, M. R., & Doiron, B. (2019). Circuit Models of Low-Dimensional Shared Variability in Cortical Networks. *Neuron*, 101(2), 337-348.e4. <https://doi.org/10.1016/j.neuron.2018.11.034>.
- Kanashiro, T., Ocker, G. K., Cohen, M. R., & Doiron, B. (2017). Attentional modulation of neuronal variability in circuit models of cortex. *ELife*, 6. <https://doi.org/10.7554/eLife.23978>.
- Lindsay, G. W., & Miller, K. D. (2018). How biological attention mechanisms improve task performance in a large-scale visual system model. *ELife*, 7, e38105. <https://doi.org/10.7554/eLife.38105>.
- Meirhaeghe, N., Sohn, H., & Jazayeri, M. (2021). A precise and adaptive neural mechanism for predictive temporal processing in the frontal cortex. *Neuron*, 109(18), 2995-3011.e5. <https://doi.org/10.1016/j.neuron.2021.08.025>.
- Mitchell, J. F., Sundberg, K. A., & Reynolds, J. H. (2009). Spatial Attention Decorrelates Intrinsic Activity Fluctuations in Macaque Area V4. *Neuron*, 63(6), 879–888. <https://doi.org/10.1016/j.neuron.2009.09.013>.
- Rabinowitz, N. C., Goris, R. L., Cohen, M., & Simoncelli, E. P. (2015). Attention stabilizes the shared gain of V4 populations. *ELife*, 4. <https://doi.org/10.7554/eLife.08998>.

Reynolds, J. H., & Heeger, D. J. (2009). The Normalization Model of Attention. *Neuron*, 61(2), 168–185. <https://doi.org/10.1016/j.neuron.2009.01.002>.

Siegle, J. H., Jia, X., Durand, S., Gale, S., Bennett, C., Graddis, N., Heller, G., Ramirez, T. K., Choi, H., Luviano, J. A., Groblewski, P. A., Ahmed, R., Arkhipov, A., Bernard, A., Billeh, Y. N., Brown, D., Buice, M. A., Cain, N., Caldejon, S., ... Koch, C. (2021). Survey of spiking in the mouse visual system reveals functional hierarchy. *Nature*, 592(7852), Article 7852. <https://doi.org/10.1038/s41586-020-03171-x>.

Steinmetz, N. A., & Moore, T. (2014). Eye Movement Preparation Modulates Neuronal Responses in Area V4 When Dissociated from Attentional Demands. *Neuron*, 83(2), 496–506. <https://doi.org/10.1016/j.neuron.2014.06.014>.

Thiele, A., & Bellgrove, M. A. (2018). Neuromodulation of Attention. *Neuron*, 97(4), 769–785. <https://doi.org/10.1016/j.neuron.2018.01.008>.

van Kempen, J., Gieselmann, M. A., Boyd, M., Steinmetz, N. A., Moore, T., Engel, T. A., & Thiele, A. (2021). Top-down coordination of local cortical state during selective attention. *Neuron*. <https://doi.org/10.1016/j.neuron.2020.12.013>.

Wang, J., Narain, D., Hosseini, E. A., & Jazayeri, M. (2018). Flexible timing by temporal scaling of cortical responses. *Nature Neuroscience*, 21(1), 102–110. <https://doi.org/10.1038/s41593-017-0028-6>.

Wasmuht, D. F., Spaak, E., Buschman, T. J., Miller, E. K., & Stokes, M. G. (2018). Intrinsic neuronal dynamics predict distinct functional roles during working memory. *Nature Communications*, 9(1), 3499. <https://doi.org/10.1038/s41467-018-05961-4>.

Zeraati, R., Engel, T. A., & Levina, A. (2022). A flexible Bayesian framework for unbiased estimation of timescales. *Nature Computational Science*, 2(3), 193–204. <https://doi.org/10.1038/s43588-022-00214-3>.

Reviewers' Comments:

Reviewer #1:

Remarks to the Author:

The authors have been responsive to the concerns of the previous round of review. Their revisions have significantly strengthened the manuscript. I support publication.

Minor comment:

This sentence at the end of par 2, page 8 felt a bit off: "The spatial selectivity of this effects suggests...may contribute to behavioral effects..." It seems equally probable (maybe more likely) that attentional modulation alters network function to improve performance and the change in slow timescales comes about as an epiphenomenon or 'biomarker' of this modulation. It seems speculative to suggest the change in timescales is contributing to behavior directly.

Reviewer #2:

Remarks to the Author:

The revised version of the manuscript has addressed issues of novelty that I brought up in the first round of review, and provided some additional elements.

A key issue have however not been resolved. Specifically, in the first round of review, I pointed out that the modulation of the timescales cannot be used to reveal connectivity structure, as this reverse inference is in general not valid. The response agrees that the original claims were too strong, yet the title still reads "Intrinsic timescales in the visual cortex ... reflect spatial connectivity", and the abstract says "Our results suggest that multiple timescales arise from the spatial connectivity in the visual cortex". These conclusions are simply not supported by the analyses in the paper.

As a minor note, the title of the Fig 4 is "Slow timescales predict behavioral performance". Yet what is shown is a *correlation*, not *prediction* and with *reaction times*, not behavioural performance as usually defined.

Reviewer #3:

Remarks to the Author:

The revision has addressed my concerns by means of the addition of new analyses and better framing of the results.

Point-by-point responses to reviewers' comments

We thank the reviewers for their constructive feedback which helped improving our manuscript. Here, we provide detailed responses to the reviewers' comments.

Reviewer #1:

The authors have been responsive to the concerns of the previous round of review. Their revisions have significantly strengthened the manuscript. I support publication.

Minor comment:

This sentence at the end of par 2, page 8 felt a bit off: "The spatial selectivity of this effects suggests...may contribute to behavioral effects..." It seems equally probable (maybe more likely) that attentional modulation alters network function to improve performance and the change in slow timescales comes about as an epiphenomenon or 'biomarker' of this modulation. It seems speculative to suggest the change in timescales is contributing to behavior directly.

We have edited this sentence to weaken the statement about possible contribution of timescales to behavioral benefits of attention:

"The spatial selectivity of this effect suggests that the increase in the slow timescale **may be related** to behavioral benefits of selective spatial attention."

Reviewer #2:

The revised version of the manuscript has addressed issues of novelty that I brought up in the first round of review, and provided some additional elements.

A key issue have however not been resolved. Specifically, in the first round of review, I pointed out that the modulation of the timescales cannot be used to reveal connectivity structure, as this reverse inference is in general not valid. The response agrees that the original claims were too strong, yet the title still reads "Intrinsic timescales in the visual cortex ... reflect spatial connectivity", and the abstract says "Our results suggest that multiple timescales arise from the spatial connectivity in the visual cortex". These conclusions are simply not supported by the analyses in the paper.

Our analyses show that timescales of cross-correlations of V4 activity depend on lateral cortical distance. The distance dependence of cross-correlations can be explained only by the network model with spatial connectivity but not by other possible local mechanisms (Fig. 5). This result supports the hypothesis that V4 timescales are related to the spatial connectivity structure. As we stated in the manuscript, a combination of multiple mechanisms (such as spatial connectivity and local biophysical processes) may jointly contribute to generating spatiotemporal correlations in visual cortex. Our results provide evidence that connectivity plays at least a partial role in defining spatiotemporal correlations in V4 and should be considered. We believe the title of the paper should reflect this finding. In the title, we used the word "reflect" (and not a stronger phrase such as "caused by" or "arise from") to clearly indicate a *suggested* relationship between timescales and connectivity and not a strong causal claim.

We edited the abstract to read: "Our results suggest that multiple timescales **may** arise from the spatial connectivity in the visual cortex" to clarify that we propose a *potential* mechanism.

As a minor note, the title of the Fig 4 is "Slow timescales predict behavioral performance". Yet what is shown is a **correlation**, not **prediction** and with **reaction times**, not behavioural performance as usually defined.

We edited the figure caption to read: "Slow timescales are correlated with monkeys' reaction times."

Reviewer #3:

The revision has addressed my concerns by means of the addition of new analyses and better framing of the results.